

# Anomalies and symmetry fractionalization

Diego Delmastro[1,2⋆], Jaume Gomis[1,2†], Po-Shen Hsin[3‡] and Zohar Komargodski[4∘]

**1** Perimeter Institute for Theoretical Physics, Waterloo, Ontario, N2L 2Y5, Canada
**2** Department of Physics, University of Waterloo, Waterloo, ON N2L 3G1, Canada
**3** Mani L. Bhaumik Institute for Theoretical Physics, 475 Portola Plaza,
Los Angeles, CA 90095, USA
**4** Simons Center for Geometry and Physics, SUNY, Stony Brook, NY 11794, USA

⋆ ddelmastro@scgp.stonybrook.edu , † jgomis@perimeterinstitute.ca ,
‡ nazgoulz@gmail.com , ∘ zkomargo@gmail.com

## Abstract

We study ordinary, zero-form symmetry $G$ and its anomalies in a system with a one-form symmetry $\Gamma$. In a theory with one-form symmetry, the action of $G$ on charged line operators is not completely determined, and additional data, a fractionalization class, needs to be specified. Distinct choices of a fractionalization class can result in different values for the anomalies of $G$ if the theory has an anomaly involving $\Gamma$. Therefore, the computation of the 't Hooft anomaly for an ordinary symmetry $G$ generally requires first discovering the one-form symmetry $\Gamma$ of the physical system. We show that the multiple values of the anomaly for $G$ can be realized by twisted gauge transformations, since twisted gauge transformations shift fractionalization classes. We illustrate these ideas in QCD theories in diverse dimensions. We successfully match the anomalies of time-reversal symmetries in $2 + 1d$ gauge theories, across the different fractionalization classes, with previous conjectures for the infrared phases of such strongly coupled theories, and also provide new checks of these proposals. We perform consistency checks of recent proposals about two-dimensional adjoint QCD and present new results about the anomaly of the axial $\mathbb{Z}_{2N}$ symmetry in $3 + 1d$ $\mathcal{N} = 1$ super-Yang-Mills. Finally, we study fractionalization classes that lead to 2-group symmetry, both in QCD-like theories, and in $2 + 1d$ $\mathbb{Z}_2$ gauge theory.

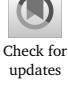

# 1 Introduction

The study of the symmetries of Quantum Field Theory (QFT) and quantum many-body systems has been an indispensable tool for several decades. Symmetries provide a robust concept that remains valid at strong coupling. As such, symmetry considerations oftentimes have inspired far-reaching insights into non-perturbative physics.

A zero-form symmetry group $G$ in a QFT comes equipped with a collection of co-dimension one topological, invertible operators which under fusion form the group $G$. The Ward identities satisfied by correlation functions of local operators transforming in representations of $G$ are a consequence of enriching the correlation functions of local operators with topological operators that implement the action of $G$.

Stronger dynamical constraints arise if the QFT has an 't Hooft anomaly [1,2] for $G$. The presence of an 't Hooft anomaly can be diagnosed by coupling to a background gauge field for the symmetry $G$. This corresponds to laying down a network of topological junctions of the topological co-dimension one operators for $G$. The symmetry $G$ has an 't Hooft anomaly if under gauge transformations of the background gauge fields for $G$ the partition function is not invariant. More precisely, the symmetry $G$ has an 't Hooft anomaly only when the non-invariance of the partition function cannot be cured by adding local counterterms for the background gauge fields. Equivalently, the 't Hooft anomaly can be described as a failure of the network of topological junctions to consistently reconnect modulo phases assigned to each junction, since gauge transformations act by recombining the network of topological junctions. The implementation of the symmetry action of $G$, and of its 't Hooft anomalies, requires defining the network of topological junctions for the symmetry $G$.

The reason that 't Hooft anomalies are interesting and powerful observables is that they are characterized by topological invariants in one dimension higher [3,4] and hence they cannot depend on continuous couplings, and also cannot evolve under the renormalization group. Indeed, 't Hooft anomalies for a symmetry $G$ admit a topological classification in terms of a cobordism group [5–19], the details of which depend on the spacetime dimension, whether the system is bosonic or fermionic, whether the symmetry acts unitarily or anti-unitarily, etc. While the early literature on 't Hooft anomalies focused mostly on the case of continuous groups $G$, it has become clear in recent years that there is a rich array of anomalies for discrete symmetries as well, and that their consequences for strong coupling dynamics are often striking.

The classification of 't Hooft anomalies for a symmetry $G$ is intrinsic. It does not depend on whether the system has additional or yet to be discovered global symmetries. Of course, the existence of additional symmetries can lead to mixed 't Hooft anomalies between $G$ and the new symmetries, but the classification of 't Hooft anomalies for $G$ is insensitive to the existence of additional global symmetries.[1]

Since anomalies are invariants of the renormalization group, they can be computed by analyzing the theory with an arbitrarily low energy cutoff. In order to compute anomalies, there is no need to discuss the degrees of freedom that may have been present at high energy. Technically, the unknown high energy degrees of freedom induce local counter-terms upon integrating them out, and as we remarked above, the anomaly is defined modulo such terms. Therefore, conventional wisdom says that anomalies are unambiguously determined given the low energy degrees of freedom and the action of the symmetry $G$ on these degrees of freedom. These properties of 't Hooft anomalies are, in part, why anomalies are so robust and useful.

In this paper we show that these general expectations about 't Hooft anomalies for $G$ need to be revised if a physical system is endowed with a one-form symmetry[2] group $\Gamma$. The study of higher symmetry and anomalies has enjoyed ample recent activity, see [24–26] for early papers on the subject. See [27–29] for reviews and additional references.

---

[1]This can fail if the zero-form symmetry $G$ "mixes" in a non-trivial fashion with a higher-form symmetry. For example, $G$ can mix with a one-form symmetry and form a two-group. Then, one should classify the two-group anomalies. It is still meaningful to discuss 't Hooft anomalies for $G$ in this case, but the two-group structure can change the classification of the $G$ anomalies. For instance, some $G$ anomalies can be cancelled by local counterterms of the two-group background gauge field and become trivial [20,21], which is the analogue of the Green-Schwarz mechanism [22] for background gauge fields. Some two-group symmetries that involve a spacetime symmetry can also allow more anomalies for the zero-form $G$ symmetry [23].

[2]More generally, any higher-form symmetry.

We will see that in the presence of one-form symmetry $\Gamma$, additional discrete data needs to be specified in order to unambiguously determine the action of the $G$ symmetry. The additional data takes values in the cohomology group $H^2_\rho(G, \Gamma)$ (see below for details). This admits several complimentary physical interpretations, which we will elucidate. Importantly, the 't Hooft anomalies for $G$ can depend on the choice of class in $H^2_\rho(G, \Gamma)$, thus yielding a spectrum of values for the 't Hooft anomalies. We will show that distinct choices of the action of $G$ can result in different 't Hooft anomalies for $G$ if the theory is endowed with an 't Hooft anomaly for $\Gamma$ or there is a mixed $G$–$\Gamma$ 't Hooft anomaly. We emphasize that this occurs regardless of any mixing of $G$ with $\Gamma$ into a higher structure, such as a 2-group or a higher generalized symmetry structure. It happens when the symmetry of the theory is the direct product $G \times \Gamma$.

There are several ways of describing the physical meaning of the additional data that needs to be given to fully specify the $G$ symmetry and its 't Hooft anomaly. In one perspective, the additional data can be ascribed to inequivalent ways of adding massive degrees of freedom that are charged under $\Gamma$ and transform in a (projective) representation of $G$. While these heavy degrees of freedom are of no direct consequence at low energies, they determine the value of the 't Hooft anomalies for $G$.[3] The reason this statement is not at odds with usual decoupling arguments is that these massive particles modify the symmetry structure of theory, and adding them changes the theory in a discontinuous fashion. Therefore, to evaluate 't Hooft anomalies for a (zero-form) symmetry $G$, one must first discover all the one-form symmetries $\Gamma$ of the theory and, subsequently, characterize all the inequivalent choices of adding massive charged particles that break the one-form symmetry.[4] This additional data can also be understood as specifying the possible ways the symmetry $G$ can act on the line operators of the theory that carry charge under $\Gamma$.[5] These two view-points are complementary, since line operators insert non-dynamical, heavy particles into the system.

Line operators can carry a projective representation under the zero-form symmetry. The projective representation is specified by the cohomology class $H^2(G_{\text{stab}}, U(1))$ of the stabilizer subgroup $G_{\text{stab}} \subset G$ that leaves the label of the line invariant [34]. In various special cases (such as conformal or topological lines), the defect Hilbert space in the presence of an insertion of a line at a point on $S^d$ is isomorphic to the space of end-points of the line defect. Then the above projective representation is realized on the space of end-points. Thus, while local operators are in a vector representation of $G$, line operators are in a projective representation of $G$. In a theory without a one-form symmetry $\Gamma$, the cohomology class of the projective representation of $G$ under which a given line operator transforms is unique and unambiguously determined.[6]

In a theory with a one-form symmetry $\Gamma$, additional choices need to be made to fully specify the action of the symmetry $G$ on the theory. That is, while the action of $G$ on local operators is unique, the action of $G$ on line operators depends on discrete choices when $\Gamma$ is non-trivial. The distinct ways that $G$ is realized on the line operators describe different $G$ "symmetry fractionalization" classes. Once the symmetry fractionalization class is fully specified, the 't Hooft anomalies for $G$ are completely fixed. We will show that distinct choices of the action of $G$ can result in different 't Hooft anomalies for $G$ if the theory is endowed with an 't Hooft

---

[3]We would like to thank T. Senthil for insightful discussions on the role of massive particles. Aspects of this appeared in [30, 31].

[4]More generally, one needs to discover all higher-form symmetries of the theory, and investigate the spectrum of extended excitations such as strings (in the case of two-form symmetry).

[5]A line operator that carries vanishing one-form charge transforms in a fixed cohomology class of projective representations of the zero-form symmetry $G$. The general statement is [32, 33] that lines not charged under one-form or non-invertible symmetries should be endable and hence admit a nonempty Hilbert space on $S^d$ whose transformation properties under the various symmetries can be studied.

[6]For example, a $2 + 1d$ $SU(2)$ gauge theory with a fermion in the **2** of $SU(2)$ has $G = SO(3)$ global symmetry. Wilson lines $W_j$ with $j \in \mathbb{Z}$ are in a vector representation of $SO(3)$ while those with $j \in 1/2 + \mathbb{Z}$ are in a projective representation of $SO(3)$. Line defects in a projective representation of the global symmetry exist also in Landau-Ginzburg models, for instance, in standard Wilson-Fisher fixed points, see [35–38] and references therein.

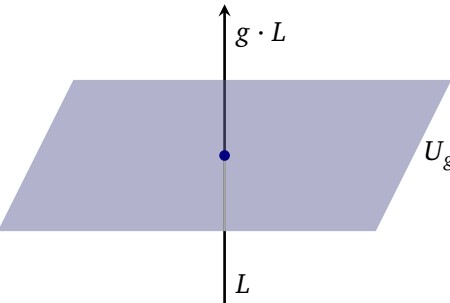

Figure 1: The type of line operator can change from $L$ to $g \cdot L$ when it pierces the codimension-one symmetry generator labelled by $g \in G$.

anomaly for $\Gamma$ or there is a mixed $G$–$\Gamma$ 't Hooft anomaly.

The phenomenon of symmetry fractionalization is well-known in the condensed matter literature for topological quantum field theories (TQFTs), see e.g. [39–48][7] and the references therein. Examples with more general quantum systems are discussed in [48, 50–52].

A simple way to understand that additional choices can be made is by considering the topological junctions for the $G$ symmetry. The topological defects in a junction intersect in co-dimension two. A one-form symmetry $\Gamma$ comes equipped with a collection of co-dimension two topological, invertible defects that fuse according to $\Gamma$. Therefore, if the theory has a one-form symmetry $\Gamma$, the junction can be decorated with a topological defect for the one-form symmetry $\Gamma$.[8] This decoration of the topological junction changes the symmetry action of $G$ on lines that carry charge under $\Gamma$, since they are acted on by the topological defect for $\Gamma$ sitting at the junction, changing the action of $G$ on charged lines by a phase. This can shift the cohomology class of projective representation of the charged lines (see figure 3). Consistency of fusion of the junctions and one-form charge implies that distinct choices of the symmetry fractionalization are related to each other by a class in[9]

$$H^2_\rho(G, \Gamma). \tag{1}$$

$\rho$ describes how elements of $G$ act on $\Gamma$

$$\rho : G \to \mathrm{Aut}(\Gamma). \tag{2}$$

The symmetry fractionalization classes depend on the automorphism $\rho$. This automorphism is fixed by the physical system, and since it acts on the line operators, it is often induced by an outer automorphism. The cohomology group describes the *jumps* in the projective representation cohomology classes that can be carried by the line operators in the theory.

Alternatively, $H^2_\rho(G, \Gamma)$ describes central extensions of $G$ by $\Gamma$. This latter interpretation of (1) admits a nice physical realization in terms of selecting a fractionalization class by inserting heavy particles that explicitly break the $\Gamma$ symmetry in the ultraviolet. These massive particles fix a certain projective representation for all the line defects in the theory. Once this is fixed, we can compute the anomaly of the zero-form symmetry $G$. The result for the anomaly of $G$ depends crucially on how the massive particles were added, naively violating the usual folklore that anomalies are insensitive to gapped degrees of freedom.

---

[7]See also [49] for the mathematical framework describing symmetry-enriched $2 + 1d$ bosonic TQFTs. The relation with one-form symmetry is discussed in [21].

[8]One may ask whether enriching junctions with non-invertible co-dimension two topological operators is possible. This would, however, be inconsistent with the fusion of junctions of a $G$ symmetry. Such a construction could give rise to a higher categorical symmetry, while the phenomena we are studying is for systems with $G \times \Gamma$ symmetry.

[9]The precise statement is that fractionalization classes form a torsor over this group. When $G$ is an extension of "bosonic symmetry" $G_b$ by the fermion parity symmetry $\mathbb{Z}_2^F$, we consider the classes given by pullback of $H^2_\rho(G_b, \Gamma)$ under the projection $G \to G_b = G/\mathbb{Z}_2^F$ [46, 47].

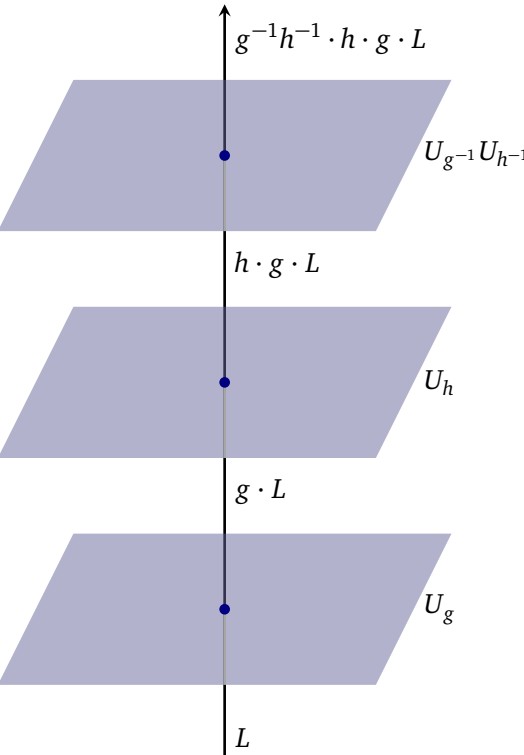

Figure 2: The collection of line operators can realize the symmetry projectively (on the Hilbert space on $S^d$), while the action of $G$ symmetry that permutes the label of lines is linear.

Consider a gauge theory with gauge group $G_{\text{gauge}}$, zero-form symmetry $G$ and one-form symmetry $\Gamma$. We add massive particles transforming under $G_{\text{gauge}}$ that explicitly break the one-form symmetry $\Gamma$. These massive particles transform in a representation of an extension of the ordinary global symmetry, which we denote by $\tilde{G}$, such that a center subgroup $\mathcal{Z} \subset \tilde{G}$ is identified with a subgroup in the center of the gauge group $G_{\text{gauge}}$. The faithful global symmetry is therefore $\tilde{G}/\mathcal{Z} = G$. The symmetry of the classical action with the massive matter fields is [53]

$$\frac{G_{\text{gauge}} \times \tilde{G}}{\mathcal{Z}}. \tag{3}$$

(We have assumed a direct product for simplicity.) Then, at low energy, the one-form symmetry of the theory without the charged massive particles emerges in the infrared, that is $\Gamma_{\text{IR}} = \Gamma = \mathcal{Z}$. The Wilson line of the massive matter fields that were added in the ultraviolet transform as a linear representation of $\tilde{G}$, which is a projective representation of $G$. Adding massive matter picks out the particular fractionalization class specified by the extension $G$ by $\Gamma$, which we denoted by $\tilde{G}$, dictated by a class in $H^2_\rho(G, \Gamma)$.

Turning on a background gauge field for $G$ that cannot be lifted to a $\tilde{G}$ gauge field, the $G_{\text{gauge}}$ gauge bundle is also modified to be a $G_{\text{gauge}}/\mathcal{Z}$ bundle, by virtue of the symmetry structure (3). This modification of the gauge bundle induced by adding the massive matter fields can contribute an 't Hooft anomaly of $G$ symmetry.[10]

---

[10]For example, consider the $SU(2)_k$ Chern-Simons theory in $2 + 1d$ with two massive scalars in the fundamental representation that breaks the one-form symmetry. Since flipping the sign of the scalar field can be identified with a gauge transformation, the faithful zero-form symmetry is $G = U(2)/\mathbb{Z}_2$. The symmetry structure is $\left(SU(2)_{\text{gauge}} \times U(2)\right)/\mathbb{Z}_2$, with $\tilde{G} = U(2)$. In the presence of a background gauge field for $G$ zero-form symmetry that cannot be lifted to a $\tilde{G}$ background gauge field, the $SU(2)_{\text{gauge}}$ gauge bundle is modified to be an $SO(3)$ bundle,

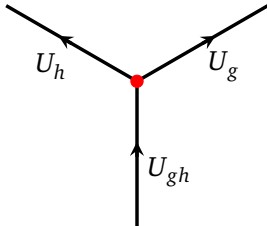

Figure 3: Junctions of co-dimension one symmetry defects are important in the study of 't Hooft anomalies. In theories with one-form symmetry, a co-dimension two invertible topological defect can be inserted in the junction.

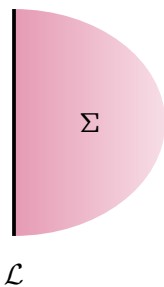

Figure 4: $1+1d$ SPT attached to line operator captures the $G$ anomaly on the line.

An elementary way of inducing a change of fractionalization class is to twist the action of $G$ with gauge transformations of $G_{\text{gauge}}$ that do not modify the $G$ symmetry algebra on the elementary fields that enjoyed the one-form symmetry (those present before adding the one-form breaking heavy fields). These gauge transformations can be thought of as de-fractionalizing the symmetry of the additional massive particles, so that they do not contribute to the 't Hooft anomalies for $G$, and the anomalies are fully captured by the original elementary fields. The twisted action can change the value of the anomaly because the charges of the elementary fields under $G$ are modified by the gauge transformation, thus giving rise to distinct values of the 't Hooft anomaly. Moreover, the twisted transformations induce a non-trivial phase that multiplies the Wilson lines of the gauge theory carrying one-form charge. This phase has a pleasing interpretation, it captures the change of projective representation of the lines under $G$, and thus these twisted transformations can be seen to induce a change of fractionalization class. The change of fractionalization class is captured by an SPT in $1+1d$ that attaches to the line and which captures the change in the 't Hooft anomaly for the $G$ symmetry on the line (see figure 4). The phase is precisely the one that arises by turning on a background two-form gauge field $B$ for the one-form symmetry $\Gamma$ taking values in $H^2_\rho(G, \Gamma)$ (see below). This way of capturing fractionalization classes will be demonstrated throughout the examples in this paper.

Instead of explicitly deforming the theory in the ultraviolet, one symmetry fractionalization class can be obtained from another one by turning on a non-trivial background two-form gauge field $B$ for the one-form symmetry $\Gamma$. We turn on a special two-form gauge field $B$ such that a co-dimension two topological operator implementing the one-form symmetry $\Gamma$ is inserted at junctions for the $G$ symmetry. The symmetry fractionalization classes are classified by

$$B = A^* \eta_2, \quad \eta_2 \in H^2_\rho(G, \Gamma), \tag{4}$$

where $A$ is the background for $G$ symmetry, and $\rho$ describes how elements of $G$ act on $\Gamma$ (2). For instance, in the gauge theory with massive matter discussed above, the symmetry structure (3)

and the Chern-Simons term for $SO(3)$ gauge bundle is not well-defined for $k \neq 0 \mod 4$. Thus the massive matter field contributes an 't Hooft anomaly for the $G$ symmetry [53].

implies that at low energies the $G_{\text{gauge}}$ gauge theory is coupled to a background two-form gauge field for the electric one-form symmetry that changes the gauge bundle to be a $G_{\text{gauge}}/\mathcal{Z}$ bundle. The fractionalization class is described by the extension $\tilde{G}$, where the extension is specified by an element $\eta_2 \in H^2_\rho(G, \mathcal{Z})$. For given $G$ background gauge field $A$, $A^*\eta_2$ is the obstruction to lifting the $G$ bundle to a $\tilde{G}$ bundle. The symmetry structure (3) implies that the background two-form gauge field equals to $A^*\eta_2$, as in (4).

Picking different fractionalization classes can yield distinct values of the 't Hooft anomaly for the zero-form symmetry $G$. Indeed, when the theory has one-form anomalies for $\Gamma$ or mixed $\Gamma$–$G$ anomalies, the topological actions (inflow terms) for these anomalies in one dimension higher can produce, upon shifting the background $B$ field for $\Gamma$ by $B \in H^2_\rho(G, \Gamma)$, a pure $G$ anomaly topological action. This is the mechanism behind different symmetry fractionalization classes resulting in different 't Hooft anomalies for $G$. In theories with one-form symmetry there is no canonical choice for $B$, as $B = 0$ is not preferred in any way (there is no canonical choice for $B$ since symmetry fractionalization classes form a torsor over $H^2_\rho(G, \Gamma)$). Instead, one has to consider all possible values of $B$ and compute the anomalies accordingly for each.

When the action of the zero-form symmetry $G$ does not commute with the action of the one-form symmetry $\Gamma$, a change of fractionalization class for the $G$ symmetry can transmute the symmetry structure of the theory from a $G \times \Gamma$ symmetry into a two-group. We demonstrate this in the $2 + 1d$ $\mathbb{Z}_2$ gauge theory for the unitary $\mathbb{Z}_2$ and time-reversal $\mathbb{Z}_2^{\mathsf{T}}$ symmetries that exchange the $e$ and $m$ particles (see appendix E). In a gauge theory, this can occur when $G$ includes charge conjugation. We show that $3 + 1d$ $\mathcal{N} = 1$ $SU(N)$ super-Yang-Mills admits, for a choice of fractionalization class, a two-group between the $G = \mathbb{Z}_2^{\mathsf{C}}$ charge conjugation symmetry and $\Gamma = \mathbb{Z}_N$ one-form symmetry (see section 5 and appendix F). In the examples we will study, when the zero-form symmetry does not involve the charge conjugation symmetry that acts on the gauge group, it does not participate in a two-group as discussed in appendix G.

Our goal in this work is twofold. First, we will go over many examples of systems with one-form symmetry and investigate the anomalies of ordinary symmetries in such circumstances. The models we will mostly discuss are quantum chromodynamics (QCD) theories: Yang-Mills theory coupled to quarks. We will devise general rules in $0+1$, $1+1$, $2+1$, and $3+1$ dimensions of how to compute the anomalies of zero-form symmetries given some data about the one-form symmetries of such theories. We will see that the ambiguity of anomalies of zero-form symmetries appears in a peculiar way, through the "non-gauge invariance" of the anomaly. While $0 + 1$ dimensional theories do not have a one-form symmetry in the usual sense, one can still discuss defects extended in time and one can still discuss projective representations associated with those defects. Therefore, examples in quantum mechanics that we study already capture some of the physics, including the fact that the choice of fractionalization class can manifest itself through the non-gauge invariance of the 't Hooft anomaly. Second, we use our improved understanding of anomalies of ordinary symmetries in the presence of one-form symmetry to test various proposed dualities for adjoint QCD (and other related theories with other quark content) both in $1 + 1$ and $2 + 1$ dimensions. We show that the zero-form anomalies, with the ambiguities due to fractionalization classes carefully taken into account, match beautifully in previously proposed scenarios for the infrared physics of adjoint QCD in $1 + 1$ and $2 + 1$ dimensions. In the latter case, the comparison of the anomalies necessitates a study of the anomalies of certain non-trivial TQFTs.

The outline of the paper is as follows. In section 2 we discuss pedagogical examples in $0 + 1$ dimensions. There is a sense of one-form symmetry in quantum mechanics, and quite many of the ideas discussed in this paper can be already demonstrated in this setting. In section 3 we consider $1 + 1$ dimensions, including bosonic quantum electrodynamics in $1 + 1d$ (QED$_2$), $\mathbb{Z}_2$ gauge theory, fermionic QED$_2$, and adjoint QCD$_2$. We discuss the anomalies of various $\mathbb{Z}_2$ global symmetries, perform consistency checks of some proposals, and devise the

rules for how to compute the anomalies of such $\mathbb{Z}_2$ symmetries properly. In section 4 we discuss time-reversal symmetry in $2+1$ dimensions. Depending on properties of the one-form symmetry, we devise the rules for how the choice of a fractionalization class influences the mod 16 time-reversal symmetry anomaly. We perform new consistency checks of proposed infrared dualities. There are important differences between $\mathsf{T}$ and $\mathsf{CT}$ which we discuss in detail and we perform consistency checks for both. In section 5 we discuss super Yang-Mills theory in $3+1$ dimensions (aka massless adjoint QCD) and compute the anomalies of its $\mathbb{Z}_{2N}$ zero-form symmetry and also show how the anomaly depends on the choice of fractionalization class. Appendices cover some technical details.

*Note added: This work is submitted in coordination with [54], which has some overlap with ours.*

## 2 Quantum mechanics

In this section we study the phenomena discussed in the introduction in an elementary setting. We consider the 't Hooft anomalies of a $0+1d$ system with $\mathbb{Z}_2^{\mathsf{T}} \times \mathbb{Z}_2^F$ zero-form symmetry, where $\mathbb{Z}_2^{\mathsf{T}}$ is generated by time-reversal $\mathsf{T}$ and $\mathbb{Z}_2^F$ by fermion parity $(-1)^F$, with $\mathsf{T}^2 = \left((-1)^F\right)^2 = 1$. 't Hooft anomalies for this symmetry are classified by the cobordism group $\Omega_2^{\mathrm{Pin}^-} = \mathbb{Z}_8$ [55]. A diagnostic of an 't Hooft anomaly in $0+1$ dimensions is that the symmetry is realized projectively on the Hilbert space. For an early study of 't Hooft anomalies in quantum mechanics see [56].

The above $\mathbb{Z}_8$ 't Hooft anomaly is captured via inflow by the $1+1d$ topological invariant

$$S = \frac{2\pi \nu}{8} \int_{\Sigma_2} ABK \,, \tag{5}$$

with $\nu \in \mathbb{Z}_8$, and $ABK$ the Arf-Brown-Kervaire invariant [10] of the surface $\Sigma_2$.

Free massless Majorana fermions in $0+1d$ have $\mathbb{Z}_2^{\mathsf{T}} \times \mathbb{Z}_2^F$ zero-form symmetry. The action of $\mathbb{Z}_2^{\mathsf{T}}$ on a Majorana fermion $\lambda$ depends on the choice of a sign

$$\mathsf{T} : \lambda(t) \to \pm \lambda(-t) \,, \tag{6}$$

while the action of $\mathbb{Z}_2^F$ is

$$(-1)^F : \lambda(t) \to -\lambda(t) \,. \tag{7}$$

Since the Hermitian mass term $i\lambda\lambda'$ is $\mathsf{T}$-invariant when $\lambda$ and $\lambda'$ transform with opposite signs under $\mathbb{Z}_2^{\mathsf{T}}$, the 't Hooft anomaly is given by

$$\nu = n_+ - n_- \mod 8 \,, \tag{8}$$

where $n_{\pm}$ is the number of Majorana fermions transforming with a $\pm$ sign under $\mathbb{Z}_2^{\mathsf{T}}$ in (6). The fact that $\nu \in \mathbb{Z}_8$ in this system follows from the existence [55] of a $\mathbb{Z}_2^{\mathsf{T}}$-symmetric quartic interaction that gaps out the fermions for $\nu = 8$.

Let us now consider QCD$_{0+1}$: gauge fields with simple and simply-connected gauge group $G_{\mathrm{gauge}}$ coupled to quarks in a representation $R$ of $G_{\mathrm{gauge}}$. QCD$_{0+1}$ is obtained by gauging a subgroup $G_{\mathrm{gauge}} \subset SO(\dim_{\mathbb{R}}(R))$ of the flavor symmetry of the free fermion theory. Gauge fields in $0+1d$ have no dynamics, but they constrain the Hilbert space and the local operators of the free fermion theory by virtue of the Gauss's law implied by the gauge field equation of motion.[11] Moreover, the gauge theory can be enriched by adding a static Wilson line stretched

---

[11]For now we ignore the difficulty with the cases of odd $\dim_{\mathbb{R}}(R)$, where there is no $\mathbb{Z}_2$-graded Hilbert space. We will take care of it later.

along time. Strictly speaking, this changes the theory, since the Wilson line insertion modifies Gauss's law, resulting in a different Hilbert space.

$\text{QCD}_{0+1}$ preserves the $\mathbb{Z}_2^{\mathsf{T}} \times \mathbb{Z}_2^F$ symmetry of the free fermion theory. Our goal is to study the $\mathbb{Z}_2^{\mathsf{T}}$ anomalies of this theory. A conventional choice for the action of $\mathbb{Z}_2^{\mathsf{T}}$ on $\text{QCD}_{0+1}$ is[12]

$$\mathsf{T}: \begin{cases} \psi(t) \to \psi^*(-t), \\ A_0(t) \to -A_0^*(-t), \end{cases} \tag{9}$$

where $\psi$ is the quark in a representation $R$ of $G_{\text{gauge}}$. Conventional wisdom would suggest that the 't Hooft anomaly for $\mathbb{Z}_2^{\mathsf{T}}$ can be be computed in the ultraviolet, where gauge interactions can be neglected, with the conclusion that the 't Hooft anomaly for the $\mathbb{Z}_2^{\mathsf{T}}$ symmetry (9) of $\text{QCD}_{0+1}$ is the number of Majorana fermions

$$\nu = \dim_{\mathbb{R}}(R) \mod 8. \tag{10}$$

This wisdom will now be scrutinized and shown to fail in $\text{QCD}_{0+1}$ theories with anomalous one-form symmetry. (We will soon explain what one-form symmetry means in quantum mechanics.)

For concreteness we consider adjoint $\text{QCD}_{0+1}$, i.e., with quarks in the adjoint representation of $G_{\text{gauge}}$. We first analyze the theory in the presence of a background gauge field for $G_{\text{gauge}}$, and discuss the effects of integrating over the gauge fields later. The Hilbert space can be straightforwardly constructed by splitting the fermions into creation and annihilation operators.[13] The dimension of the Hilbert space is $2^{\dim G_{\text{gauge}}/2}$. These states furnish the two complex spinor representations of $Spin(\dim G)$; one spinor represents the bosonic states and the other the fermionic states.

We now consider the partition function on a circle with anti-periodic boundary conditions in the presence of a background gauge field $a$, which can be brought to the Cartan subalgebra of $\mathfrak{g}_{\text{gauge}}$ by a gauge transformation. Either by computing the fermion determinant, or by evaluating the Hamiltonian on the Hilbert space, we find that

$$Z_{\text{NS}}(a) = 2^{\text{rank}(G)/2} \prod_{\alpha > 0} \left( e^{i\pi\alpha(a)} + e^{-i\pi\alpha(a)} \right), \tag{11}$$

where the product runs over the positive roots $\alpha$ of the Lie algebra of $G_{\text{gauge}}$. The residual gauge transformations after gauge fixing are the time independent gauge transformations, which act by conjugation. This implies that the partition function is a class function of $U = e^{2\pi i a}$, i.e., it must admit a decomposition into characters of $G_{\text{gauge}}$.

A computation shows that the partition function of adjoint $\text{QCD}_{0+1}$ can be expressed as the character in the representation of $G_{\text{gauge}}$ with highest weight the Weyl vector $\rho = (1, 1, \ldots, 1)$. It is given by

$$Z_{\text{NS}} = 2^{\text{rank}(G_{\text{gauge}})/2} \chi_{(1,1,\ldots,1)}(U), \tag{12}$$

where

$$\chi_{(1,1,\ldots,1)}(U) = W_\rho \equiv \text{tr}_\rho \, \text{P} \exp\left( i \oint a \right). \tag{13}$$

Therefore, the partition function can be described as the insertion of a Wilson line $W_\rho$ in the representation $\rho$ of $G$ in the theory without fermions. So far $a$ was a fixed Lie algebra element.

---

[12]Other time-reversal symmetries can be defined by combining $\mathsf{T}$ with a unitary global $\mathbb{Z}_2$ symmetry of the $\text{QCD}_{0+1}$ theory.

[13]In order to simplify the discussion, we add an uncharged Majorana fermion when $\dim G_{\text{gauge}}$ is odd so that the theory is endowed with a conventional graded Hilbert space. Whenever we write $\dim G_{\text{gauge}}$ we refer to the setup where the total number of fermions is even, and includes the case where we may have added one decoupled Majorana fermion to make the total number of fermions even.

For $G_{\text{gauge}} = SU(N)$, the theory has a faithful $PSU(N)$ global symmetry acting on the quarks $\psi$ in the adjoint representation, but on the Hilbert space, the symmetry is realized projectively corresponding to a $1+1d$ SPT phase for the $PSU(N)$ global symmetry with coefficient $N/2$ mod $N$ for even $N$ and $0$ mod $N$ otherwise. A similar discussion holds for all simple and simply-connected $G_{\text{gauge}}$.

We now proceed to gauge the $G_{\text{gauge}}$ symmetry. This requires path integrating over the gauge fields, i.e. carefully summing over all group elements $\int [DU]$ with $[DU]$ the Haar measure on $G_{\text{gauge}}$.[14] The theory we obtain in this way is adjoint $\text{QCD}_{0+1}$.

The presentation of the partition function in (12) makes it clear that the Hilbert space of adjoint $\text{QCD}_{0+1}$ is empty. This is simply because integrating the character $\chi_{(1,1,\dots,1)}(U)$ we find a vanishing partition function due to the orthogonality of characters

$$\int [DU] \chi_{R_1}(U) \chi_{R_2}(U^*) = \delta_{R_1,R_2}, \tag{14}$$

that is $Z_{\text{QCD}_{0+1}} = 0$. The Gauss's law constraint eliminates all the states and the physical Hilbert space is empty.

## 2.1 One-form symmetry

It is useful to think about these cases where the partition function vanishes as due to a one-form symmetry with an 't Hooft anomaly, in a sense we now explain. Consider the following Hilbert space

$$\mathcal{H} = \bigoplus_{r \in Z[G_{\text{gauge}}]^\vee} \mathcal{H}_r, \tag{15}$$

where $Z[G_{\text{gauge}}]$ is the center of $G_{\text{gauge}}$ (see table 1). A state in $\mathcal{H}_r$ carries charge $r \in Z[G_{\text{gauge}}]^\vee$. Therefore on $\mathcal{H}_r$ the zero-form symmetry $G_{\text{gauge}}$ is realized with an anomaly since it acts projectively. The same coefficient $r \in Z[G_{\text{gauge}}]^\vee$ will be soon interpreted as a one-form symmetry anomaly after gauging $G_{\text{gauge}}$. The operators that interpolate between Hilbert spaces with different charges are line operators charged under $Z[G_{\text{gauge}}]$.[15] Of course, in $0+1d$ inserting a Wilson line changes the theory, so that the Hilbert space above can be regarded as the Hilbert space of decoupled $0+1d$ theories. It is in this sense that it is meaningful to assign one-form charge $r \in Z[G_{\text{gauge}}]^\vee$ to the partition function of $\mathcal{H}_r$ after gauging $G_{\text{gauge}}$, and the assigned charge is compatible with the fusion of line operators. The one-form charge $r \in Z[G_{\text{gauge}}]^\vee$ will be now interpreted as the one-form symmetry anomaly. Whenever the one-form symmetry anomaly is nonzero, the corresponding theory has a vanishing partition function because the one-form symmetry charge allows to rotate the partition function by a phase which is some non-trivial root of unity (and the only partition function that may be compatible with that is the vanishing one).[16]

An invertible theory of a two-form gauge field in $1+1d$ captures via inflow the one-form anomalies of the boundary $0+1d$ theory. One-form anomalies for a $\mathbb{Z}_M$ one-form symmetry are classified by $\Omega_2^{Spin}(B^2\mathbb{Z}_M) = \mathbb{Z}_2 \times \mathbb{Z}_M$. The first factor corresponds to the Arf invariant and the second to

$$S = \frac{2\pi k}{M} \int_{\Sigma_2} B \qquad k = 0, 1, \dots, M-1. \tag{16}$$

---

[14]For details how the integration measure over $\mathfrak{g}_{\text{gauge}}$ transmutes to the Haar measure of $G_{\text{gauge}}$ upon gauge fixing, see e.g. Appendix $C$ in [57].

[15]This resembles what happens in $1+1d$, where Hilbert spaces graded by different one-form charge are decoupled and can only be interpolated by acting with a line operator that is charged under the one-form symmetry.

[16]The Hilbert space might be empty also in the absence of one-form anomaly, e.g., for $SU(N)$ with $N$ odd, the theory has one-form symmetry $\mathbb{Z}_N$, there is no anomaly, and yet $Z_{\text{QCD}} = 0$.

| $G_{\text{gauge}}$ | $SU(N)$ | $Sp(N)$ | $Spin(2N+1)$ | $Spin(4N)$ | $Spin(4N+2)$ | $E_6$ | $E_7$ | $E_8$ | $F_4$ | $G_2$ |
|---|---|---|---|---|---|---|---|---|---|---|
| $Z(G_{\text{gauge}})$ | $\mathbb{Z}_N$ | $\mathbb{Z}_2$ | $\mathbb{Z}_2$ | $\mathbb{Z}_2 \times \mathbb{Z}_2$ | $\mathbb{Z}_4$ | $\mathbb{Z}_3$ | $\mathbb{Z}_2$ | $\cdot$ | $\cdot$ | $\cdot$ |

Table 1: The centers of simple Lie Groups.

The $\mathbb{Z}_M$ factor simply encodes the charge of the Hilbert space under the one-form symmetry.

If we impose $\mathbb{Z}_2^{\mathsf{T}}$ time-reversal symmetry, anomalies are classified instead by $\Omega_2^{Pin^-}(B^2\mathbb{Z}_M) = \mathbb{Z}_8 \times \mathbb{Z}_{\gcd(M,2)}$. The first factor is the pure $\mathbb{Z}_2^{\mathsf{T}}$ anomaly and is described by (5), while the one-form anomaly in a system with a $\mathbb{Z}_2^{\mathsf{T}}$ symmetry reduces to $\mathbb{Z}_{\gcd(M,2)}$. This can be understood from the action (16) being invariant under $\mathsf{T}$ only for $k = 0, M/2$, so there is no anomaly for $M$ odd and it is $\mathbb{Z}_2$ for $M$ even. Therefore, the 't Hooft anomalies of a $0+1d$ system with a $\mathbb{Z}_2^{\mathsf{T}} \times \mathbb{Z}_2^F$ zero-form symmetry and a $\mathbb{Z}_M$ one-form symmetry are

$$S_{Pin^-} = \frac{2\pi\nu}{8}\int_{\Sigma_2} ABK + \pi\mu \int_{\Sigma_2} B \qquad M \text{ even},$$

$$S_{Pin^-} = \frac{2\pi\nu}{8}\int_{\Sigma_2} ABK \qquad\qquad M \text{ odd}, \tag{17}$$

where $\nu \in \mathbb{Z}_8$ and $\mu \in \mathbb{Z}_2$.

We can readily compute the anomaly. It follows from the partition function of adjoint $\text{QCD}_{0+1}$ (12) that the one-form charge of the vacuum is

$$(1, 1, \ldots, 1) \cdot \nu \quad \text{mod } |Z(G_{\text{gauge}})|, \tag{18}$$

where $\nu$ is the congruence vector of $G_{\text{gauge}}$.[17] Since our adjoint $\text{QCD}_{0+1}$ theories enjoy time-reversal symmetry in the absence of line operators insertions, the one-form anomaly is valued in $\mathbb{Z}_2$. Further, the anomaly vanishes when the generator of the one-form symmetry has odd order.

In appendix A we evaluate $(1, 1, \ldots, 1) \cdot \nu \mod |Z(G_{\text{gauge}})|$ for all the classical Lie groups. The result is as follows: the adjoint $\text{QCD}_{0+1}$ theories that have a one-form anomaly are

| $N \bmod 4$ | 0 | 1 | 2 | 3 |
|---|---|---|---|---|
| $SU(N)$ | ✓ | | ✓ | |
| $Sp(N)$ | | ✓ | ✓ | |
| $Spin(2N+1)$ | ✓ | ✓ | ✓ | ✓ |
| $Spin(2N)$ | | | ✓ | ✓ |

(19)

Of the remaining simple Lie groups that have a center we find that $E_7$ adjoint $\text{QCD}_{0+1}$ has a one-form 't Hooft anomaly while the $E_6$ theory does not.

One may diagnose the one-form 't Hooft anomaly by attempting to gauge it. This corresponds to studying $\text{QCD}_{0+1}$ with gauge group $G_{\text{gauge}}/\Gamma$. If $\Gamma$ has a 't Hooft anomaly, then the theory with gauge group $G_{\text{gauge}}/\Gamma$ suffers from a global gauge anomaly, and it is an inconsistent model: the theory is not invariant under large $G_{\text{gauge}}/\Gamma$ gauge transformations (see e.g. [56]).

The theories with the one-form anomaly have an empty Hilbert space, but they can be enriched so that the Hilbert space is non empty by considering a Wilson line insertion $W_\rho$ in the representation of $G_{\text{gauge}}$ with highest weight given by the Weyl vector $\rho = (1, 1, \ldots, 1)$.[18] We will denote this Hilbert space by $\mathcal{H}_\rho$.

---

[17]By definition, the expression $\lambda \cdot \nu \mod |Z(G_{\text{gauge}})|$ determines the charge under the center of the representation of $G_{\text{gauge}}$ with highest weight $\lambda$. See appendix A for the value of $\nu$ for the simple Lie groups.

[18]The insertion of any other Wilson line would lead to an empty Hilbert space by the orthogonality of characters.

## 2.2 Time-reversal symmetry

We proceed now to determine the $\mathbb{Z}_2^{\mathsf{T}}$ time-reversal anomaly by analyzing how $\mathsf{T}$ is realized on the Hilbert space $\mathcal{H}_\rho$. We recall that local operators are in a (vector) representation of $\mathbb{Z}_2^{\mathsf{T}}$, but line operators can be in a projective representation of $\mathbb{Z}_2^{\mathsf{T}}$. For a line operator with one-form charge, the projective representation carried by the line takes values in $H^2(\mathbb{Z}_2^{\mathsf{T}}, \mathbb{Z}_M) = \mathbb{Z}_{\gcd(2,M)}$, where $\mathbb{Z}_M$ is the one-form symmetry. The line $W_\rho$ in adjoint $\mathrm{QCD}_{0+1}$ carries one-form charge precisely when the theory has a one-form anomaly (see (19)). This means that in the adjoint $\mathrm{QCD}_{0+1}$ theories with no 't Hooft anomaly for the one-form symmetry, the action of $\mathbb{Z}_2^{\mathsf{T}}$ on the local and line operators is completely fixed. This, in turn, implies that for these theories the $\mathbb{Z}_2^{\mathsf{T}}$ 't Hooft anomaly takes a fixed value in $\mathbb{Z}_8$. The $\mathbb{Z}_2^{\mathsf{T}}$ 't Hooft anomaly of the adjoint $\mathrm{QCD}_{0+1}$ theories that have one-form symmetry with anomaly require picking an element $H^2(\mathbb{Z}_2^{\mathsf{T}}, \mathbb{Z}_M) = \mathbb{Z}_2$, which corresponds to fixing the action of $\mathbb{Z}_2^{\mathsf{T}}$ on $W_\rho$ (and all charged lines). Indeed, $W_\rho$ can be either in a Kramers singlet *or* Kramers doublet representation of $\mathbb{Z}_2^{\mathsf{T}}$, and as we will show the $\mathbb{Z}_2^{\mathsf{T}}$ 't Hooft anomaly depends crucially on this choice.

The action of $\mathsf{T}$ when $W_\rho$ is a Kramers singlet is

$$\mathsf{T}|W_\rho\rangle = \prod_{n=1}^{\frac{1}{2}\dim G_{\text{gauge}}} \psi_n^\dagger |W_\rho\rangle. \tag{20}$$

Time-reversal acts as particle-hole symmetry. This is to be contrasted with the action of time-reversal when $W_\rho$ is in a Kramers doublet

$$\hat{\mathsf{T}}|W_\rho\rangle_+ = \prod_{n=1}^{\frac{1}{2}\dim G_{\text{gauge}}} \psi_n^\dagger |W_\rho\rangle_-,$$

$$\hat{\mathsf{T}}|W_\rho\rangle_- = - \prod_{n=1}^{\frac{1}{2}\dim G_{\text{gauge}}} \psi_n^\dagger |W_\rho\rangle_+. \tag{21}$$

Let us now recall [58] that the $\mathbb{Z}_2^{\mathsf{T}}$ anomaly $\nu \in \mathbb{Z}_8$ can be computed by determining whether $\mathsf{T}$ is fermion even or odd on the Hilbert space and whether $\mathsf{T}^2 = 1$ or $\mathsf{T}^2 = -1$ on the Hilbert space. While the fermion parity of $\mathsf{T}$ and $\hat{\mathsf{T}}$ is clearly identical, the way the $\mathbb{Z}_2^{\mathsf{T}}$ symmetry algebra is realized on $\mathcal{H}_\rho$ differs by a sign

$$\hat{\mathsf{T}}^2 = -\mathsf{T}^2. \tag{22}$$

This implies that the time-reversal anomaly for the two choices of action of time-reversal on $W_\rho$ differ by 4

$$\nu_{\hat{\mathsf{T}}} = \nu_{\mathsf{T}} + 4. \tag{23}$$

This elementary example exposes a ubiquitous phenomenon: the 't Hooft anomalies of a zero-form symmetry can depend on the choice of projective representation of the zero-form symmetry on the line operators.

We next describe three equivalent ways to understand the result (23). The first one consists of exploring the anomaly polynomial of the theory; the second one of adding one-form symmetry breaking heavy scalars; and the third one of studying how Wilson lines transform under suitable shifts of the gauge connection.

Anomaly polynomial

The fact that the anomaly of the two implementations of time-reversal differ by $\Delta \nu = 4$ is a consequence of the one-form anomaly of the theory. The anomaly for the one-form symmetry in the presence of time-reversal is

$$\pi \int_{\Sigma_2} B \,. \tag{24}$$

The change of action of time-reversal on the lines is implemented by turning a fractionalization class $B \in H^2(\mathbb{Z}_2^{\mathsf{T}}, \mathbb{Z}_N)$. This corresponds to setting $B = w_1^2$. The $\mathbb{Z}_2^{\mathsf{T}}$ anomaly therefore shifts by

$$\pi \int w_1^2 \,. \tag{25}$$

This induces a shift of $\Delta \nu = 4$ by virtue of the identity

$$\pi \int w_1^2 = \pi \int ABK \quad \mathrm{mod}\ 2\mathbb{Z} \,. \tag{26}$$

This nicely realizes the different anomalies as arising from distinct symmetry fractionalizations.

### One-form symmetry breaking heavy particles

We will now show an additional derivation of the 4 mod 8 ambiguity in the time-reversal anomaly due to the one-form symmetry anomaly. We will break the one-form symmetry explicitly by adding massive scalar particles $\phi$. The anomaly is sensitive to such massive particles and we will see that distinct ways of adding such massive particles result in a 4 mod 8 freedom in the time-reversal anomaly (23). The sensitivity of the anomaly to these massive particles can be established in several ways, as we will see. In particular, one way concerns with composing $\mathsf{T}$ with various gauge transformations, which we will show directly leads to the fractionalization classes $B \in H^2(\mathbb{Z}_2^{\mathsf{T}}, \mathbb{Z}_N)$.

The Wilson lines from before are now interpreted as the wordline of these massive particles; as such, the symmetry fractionalization is now detected as a fractionalized action on the massive scalars.

The most general anti-unitary transformation acting on the massless fermions $\psi$ and the new massive scalars $\phi$ in the fundamental representation of $G_{\mathrm{gauge}}$ is

$$\begin{aligned} \mathsf{T}(\psi) &= (R(U) \cdot \psi)^* \\ \mathsf{T}(\phi) &= (U\phi)^* \,, \end{aligned} \tag{27}$$

where $U \in G_{\mathrm{gauge}}$ is some matrix in color space, and $R$ is the representation under which the fermions transform. Of course our main interest in this section is in the case when the fermions are in the adjoint representation, but we carry out the analysis for general $R$ below. The transformation (27) is a symmetry of $QCD_{0+1}$ provided we transform the gauge field as $A_0 \mapsto -(UA_0 U^{-1})^*$.

We want to choose $U$ such that the action of time-reversal on $\psi$ is not fractionalized, namely we impose that

$$\mathsf{T}^2(\psi) = \psi \,. \tag{28}$$

This requires that $R(U^2) = 1$ when acting on the fermions, i.e., that $U^2 \in \ker(R)$. The fractionalized action of $\mathsf{T}$ is entirely carried by the scalars

$$\mathsf{T}^2(\phi) = U^2 \phi \,. \tag{29}$$

Of course, the time-reversal algebra is still $\mathsf{T}^2 = 1$, since $U^2$ is a gauge transformation. But as long as $U^2 \neq 1$, time-reversal will act projectively on the scalars.

Note that, for non-trivial $R$, the kernel $\ker(R)$ is always a subgroup of the center of the gauge group. In particular, it is the one-form symmetry of the original theory before adding the scalars: it is the subgroup of $Z(G_{\text{gauge}})$ that cannot be screened by the fermions. Therefore, $U^2$ is always a central element, i.e., $\mathsf{T}^2$ acts on $\phi$ as a phase, a $|\ker(R)|$ root of unity. Choosing different fractionalization classes amounts to making a choice for this phase. As we scan for different matrices $U \in G_{\text{gauge}}$, we realize the different projective representations of $\mathsf{T}$ acting on the massive scalars, i.e., on the Wilson lines.

As the action of $\mathsf{T}$ on $\psi$ is not fractionalized, computing the time-reversal anomaly is straightforward: we only have to count how many components of $\psi$ transform as $+1$ under $R(U)$, and how many as $-1$. Note that, since $R(U)$ has eigenvalues $\pm 1$, the anomaly is easily calculated as $\nu = \operatorname{tr} R(U) := \operatorname{tr}_R(U)$. This is invariant under conjugation and thus we can assume without loss of generality that $U$ sits in some maximal torus of $G_{\text{gauge}}$.

Let us now quote the results for the adjoint representation $R = \text{adj}$, which acts as $R(U) \cdot \psi = U \psi U^{-1}$. Clearly, $\ker(\text{adj}) = Z(G_{\text{gauge}})$, and the one-form symmetry is the center of the gauge group. In the orthogonal case we make no distinction between $Spin(N)$ and $SO(N)$, since the central $\mathbb{Z}_2$ does not act on the adjoint representation.

**Unitary group $SU(N)$.** The most general diagonal matrix $U \in SU(N)$ that satisfies $R(U^2) = 1$ is $U \propto \operatorname{diag}(-1_p, +1_{N-p})$ for some integer $p$. The associated anomaly is

$$\nu_p = \operatorname{tr}(U)\operatorname{tr}(U^{-1}) - 1 = (N - 2p)^2 - 1. \tag{30}$$

This is invariant modulo 8 under $p \to p + 2$, and therefore it is enough to look at $p = 0, 1$. When $N$ is odd, we find $\nu_0 = \nu_1 \mod 8$, and therefore $\Delta \nu = 0$. When $N$ is even, we find $\nu_0 = \nu_1 + 4$, and therefore $\Delta \nu = 4$.

**Symplectic group $Sp(N)$.** The most general diagonal matrix $U \in Sp(N)$ that satisfies $R(U^2) = 1$ is either $U = \operatorname{diag}(-1_{2p}, +1_{N-2p})$ for some integer $p$, or $U' = i \operatorname{diag}(-1_N, +1_N)$. The associated anomalies are

$$\nu_p = \frac{1}{2}(\operatorname{tr}(U^2) + \operatorname{tr}(U)^2) = \frac{1}{2}(2N + (2N - 4p)^2) \tag{31}$$

and

$$\nu' = \frac{1}{2}(\operatorname{tr}(U'^2) + \operatorname{tr}(U')^2) = -N. \tag{32}$$

Note that $\nu_p$ is independent of $p$ modulo 8. The only non-trivial shift is $\Delta \nu = \nu_0 - \nu' = 2N(N+1)$, which equals 4 when $N = 1, 2 \mod 4$, and vanishes otherwise.

**Orthogonal group $SO(N)$.** The most general matrix $U$ in a maximal torus of $SO(N)$ that satisfies $R(U^2) = 1$ is either $U = \operatorname{diag}(-1_{2p}, +1_{N-2p})$ for some integer $p$, or $U' = \operatorname{diag}(i\sigma_y, \ldots, i\sigma_y)$ if $N$ is even. The associated anomalies are

$$\nu_p = \frac{1}{2}(-\operatorname{tr}(U^2) + \operatorname{tr}(U)^2) = \frac{1}{2}(-N + (N - 4p)^2), \tag{33}$$

and

$$\nu' = \frac{1}{2}(-\operatorname{tr}(U'^2) + \operatorname{tr}(U')^2) = \frac{1}{2}N. \tag{34}$$

As before, $\nu_p$ is invariant modulo 8 under $p \to p + 2$, and therefore it is enough to consider $p = 0, 1$. When $N$ is even we have $\nu_0 = \nu_1 \mod 8$, while for odd $N$ we have $\nu_0 = \nu_1 + 4 \mod 8$. On the other hand, $\nu_0 - \nu'$ equals 4 when $N = 4, 6 \mod 8$.

To summarize this discussion, the shifts are $\Delta \nu = 4$ when

| $N \bmod 4$ | 0 | 1 | 2 | 3 |
|---|---|---|---|---|
| $SU(N)$ | ✓ | | ✓ | |
| $Sp(N)$ | | ✓ | ✓ | |
| $Spin(2N+1)$ | ✓ | ✓ | ✓ | ✓ |
| $Spin(2N)$ | | | ✓ | ✓ |

(35)

which exactly matches our previous calculation (19).

### Action on Wilson lines

In the preceding discussion we have seen that the various fractionalization classes and anomaly jumps can be explored by combining $\mathsf{T}$ with gauge transformations. Let us explain from another point of view why this is the case. For simplicity we assume that the one-form symmetry group is cyclic.

In the presence of background gauge field $w_1$ for $\mathsf{T}U$, with $U \in G_{\text{gauge}}$, the gauge connection is shifted by $w_1$ as

$$A \to A + \mathfrak{u} w_1 \,, \tag{36}$$

where $U = e^{i\mathfrak{u}}$.

Let us consider the action of $\mathsf{T}U$ on the Wilson line operators $W_R$, labeled by a representation $R$ of $G_{\text{gauge}}$. Due to the shift of the gauge field (36), in the presence of background $w_1$, the Wilson line changes as

$$\begin{aligned}
\text{tr}_R(e^{i\int A}) &\mapsto \text{tr}_R(U^{\int w_1} e^{i\int A}) \\
&= \text{tr}_R(U^{2\int_\Sigma dw_1/2} e^{i\int A})\,,
\end{aligned} \tag{37}$$

where $\Sigma$ is the surface that bounds the line (see figure 4).

As discussed in the previous subsection, $\mathsf{T}U$ will act non-projectively on the fundamental fermions $\psi$ if and only if $U^2 \in \ker(R) \equiv \Gamma$, i.e., if $U^2$ is an element of the one-form symmetry group $\Gamma$. This implies that $U^2$ is central, say

$$U^2 = e^{2\pi i \frac{\kappa}{|\Gamma|}} \mathbf{1}\,, \tag{38}$$

for some integer $\kappa \in \Gamma^\vee$. Plugging this into (37) we conclude that, under (36), Wilson lines transform as

$$W_R \mapsto e^{2\pi i \frac{\kappa z_R}{|\Gamma|} \int_\Sigma \frac{1}{2} dw_1} W_R\,, \tag{39}$$

where $z_R$ is the charge of $R$ under $\Gamma$.

On the other hand, a Wilson line with one-form charge $z_R \in \Gamma^\vee$ couples to a background field for $\Gamma$ as

$$e^{2\pi i \frac{z_R}{|\Gamma|} \int_\Sigma B} W_R\,. \tag{40}$$

Therefore, by combining time-reversal symmetry with the gauge transformation $U$, we have induced the following background field

$$B = \kappa \frac{dw_1}{2}\,. \tag{41}$$

When $\kappa$ is even, or when $|\Gamma|$ is odd, $B$ is pure gauge and cohomologically trivial, thus not resulting in a shift of the $\mathbb{Z}_2^\mathsf{T}$ anomaly. Instead, for even $|\Gamma|$ and odd $\kappa$, the shift induces a non-trivial fractionalization class.[19]

---

[19]We remark that changing the fractionalization classes of symmetry $G$ by shifting the background $B$ field for the one-form symmetry $\Gamma$ can be viewed as an outer automorphism for the two-group that combines the one-form and ordinary symmetries (including the split two-group that factorizes into a semidirect product of these symmetries). When the two-group factorizes, such outer automorphism acts trivially on the one-form symmetry and projects to the identity on $G$, as classified by the fractionalization classes $H_\rho^2(G, \Gamma)$. The outer automorphism in general also shifts the 't Hooft anomaly.

As an illustration, in $SU(N)$ we argued that the matrices that resulted in a non-fractionalized action on the fundamental fermions were $U = e^{i\pi p/N} \text{diag}(-1_p, 1_{N-p})$. For these matrices we find $U^2 = e^{2i\pi p/N} 1_N$, i.e., $\kappa \equiv p$. We see that $p dw_1/2$ is a coboundary for any $N, p$ unless $N$ is even and $p$ is odd, which precisely matches our computation in the previous subsection (cf. the discussion below (30)).

We remark that the manipulations in this subsection are correct for the time-reversal symmetry $\mathsf{T}$ but they fail for $\mathsf{CT}$ because the gauge symmetry and $\mathsf{CT}$ do not commute. We shall return to $\mathsf{CT}$ presently.

Fractionalization classes and twisted gauge bundles

Another way to see the gauge bundle is modified is by computing the magnetic fluxes. Let us consider the $SU(N)$ gauge theory example above. If we change $\mathsf{T}$ to $\mathsf{T}U$ and turn on a non-trivial background for the time-reversal symmetry, i.e. place the theory on an unorientable $pin^-$ manifold, we will see that the sum over the gauge bundles is modified. If we consider gauge field in the Cartan torus, $A = (a_1, \cdots, a_{N-1}, -(a_1 + \cdots + a_{N-1}))$, the Stiefel-Whitney class is

$$w_2^{(N)} = N \sum_{i=1}^{N-1} \frac{da_i}{2\pi} \mod N. \tag{42}$$

For ordinary $SU(N)$ gauge field, $a_i$ are properly quantized and the Stiefel-Whitney class is always trivial. In the presence of background $B$ for the $\mathbb{Z}_N$ one-form symmetry, the bundle is modified to $PSU(N)$ bundle with $w_2^{(N)} = B$.

The shift of the gauge field (36) changes the sum of GNO fluxes to be

$$w_2^{(N)} = N \frac{1}{2\pi} \left( -\pi \frac{p}{N} dw_1 \right) = -p \frac{dw_1}{2} \mod N. \tag{43}$$

Thus this is equivalent to shifting the $B$ field as

$$B \to B - p \frac{dw_1}{2} \mod N. \tag{44}$$

When $p$ is even, $p dw_1/2 = (p/2)dw_1$ is exact, and this a trivial shift in the $B$ field. For odd $N$, the shift is exact for every $p$, since 2 is invertible in $\mathbb{Z}_N$. For even $N$ and odd $p$, this is a non-trivial change in the background $B$.

In particular, this implies that the line operators that transform under the one-form symmetry with odd charge carry a projective representation of the time-reversal symmetry $\mathsf{T}U$. Under a background gauge transformation of time-reversal symmetry

$$w_1 \to w_1 + d\lambda_0 + 2\lambda_1, \quad B \to B - p d\lambda_1, \tag{45}$$

where we take a lift of $w_1$ to an integer one-cochain and different lifts are related by $2\lambda_1$ for integer one-cochain $\lambda_1$. Thus the time-reversal transformation induces a one-form transformation, which acts on the line operators. We note that the above discussion about the shift in $B$ background gauge field works in general spacetime dimension.

## 2.3 Anomalies and fractionalization in $\mathsf{CT}$

It is also interesting to perform the same analysis for $\mathsf{CT}$. This is a distinct symmetry from $\mathsf{T}$ whenever the gauge group admits non-trivial outer automorphisms, i.e., when the Dynkin diagram has a reflection symmetry. This is so for the ADE algebras. For concreteness we focus on $SU(N)$. We follow the strategy of adding massive fundamental scalars in order to detect the fractionalized action on the Wilson lines.

For complex representations, $\nu_{\mathsf{CT}} = 0$. This follows from the fact that the real and imaginary parts of $\psi$ transform with opposite sign; equivalently, one can write down a CT-symmetric mass term $\bar{\psi}\psi$. Therefore, we only expect non-trivial $\nu_{\mathsf{CT}}$ for real representations (where the mass term above vanishes by fermi statistics, since the singlet in $R \otimes R$ is symmetric). We consider the adjoint representation.

In adjoint $SU(N)$ enriched with massive fundamental scalars, CT acts as

$$
\begin{aligned}
\mathsf{CT}(\psi) &= U\psi U^{-1}\,, \\
\mathsf{CT}(\phi) &= U\phi\,,
\end{aligned}
\tag{46}
$$

where $U \in SU(N)$ is some matrix in color space. In the case of T we argued that we could always conjugate $U$ to a maximal torus of the gauge group. The reason is that change of bases acted as $U \mapsto VUV^{-1}$, and this can always be used to diagonalize $U$. On the other hand, for CT, change of bases act as $U \mapsto V^*UV^{-1}$, and this cannot always be used to diagonalize $U$. Therefore, here we cannot assume that $U$ lies in some maximal torus of $SU(N)$. We need to work a little harder.[20]

To begin with, we require CT to act non-projectively on the fermions, $\mathsf{CT}^2(\psi) = \psi$, which yields the condition

$$
(U^*U)\psi(U^*U)^{-1} = \psi\,.
\tag{49}
$$

This means that $U^*U$ must commute with $\psi$, i.e., it must be a multiple of the identity. Therefore, $U^t = cU$ for some constant $c$; taking the transpose of this equation and plugging it back into itself we derive the condition $U = c^2 U$, namely $U$ must satisfy $U^t = \pm U$.

In conclusion, the most general matrix $U \in SU(N)$ such that CT acts non-projectively on $\psi$ is either a symmetric matrix or an anti-symmetric matrix. For odd $N$ the latter option is incompatible with $\det(U) = 1$. The action of such a matrix on the scalars becomes

$$
\mathsf{CT}^2(\phi) = U^*U\phi \equiv \pm\phi\,,
\tag{50}
$$

when $U^t = \pm U$. We thus see that a symmetric matrix $U$ does not lead to a fractionalized action on $\phi$, while an anti-symmetric matrix does. Moreover, the possible projective actions are only a sign $\pm$, as opposed to an arbitrary $N$-th root of unity as was the case for T.

We now proceed to compute the anomaly $\nu_{\mathsf{CT}}$. We exploit the freedom to change bases $U \mapsto V^*UV^{-1}$ to bring $U$ into a canonical form. Note that this redefinition preserves whether $U$ is symmetric or anti-symmetric. Using this change of basis, we can always bring any unitary matrix into the following form:

$$
\begin{aligned}
U &\propto 1\,, & &\text{if } U \text{ is symmetric}\,, \\
U &\propto \mathrm{diag}(i\sigma_y,\ldots,i\sigma_y)\,, & &\text{if } U \text{ is anti-symmetric}\,.
\end{aligned}
\tag{51}
$$

---

[20]If we use a diagonal transformation $U$, the gauge bundle is not twisted by a quotient for CT$U$.

Since the time-reversal action includes the charge conjugation C, let us first review the properties of the bundle $SU(N) \rtimes \mathbb{Z}_2^{\mathsf{C}}$. In the presence of background $B_1$ for the charge conjugation symmetry, the flux quantization for $SU(N) \rtimes \mathbb{Z}_2^{\mathsf{C}}$ bundle is instead (see e.g. [59])

$$
N\sum_{i=1}^{N-1}\frac{D_{B_1}a_i}{2\pi} = 0 \bmod N\,,
\tag{47}
$$

where the covariant derivative can be unpacked by choosing a gauge and embedding $B_1$ instead of a $U(1)$ gauge field, as $D_{B_1}a_i = da_i - 2B_1 a_i$ [59].

In our case, $B_1 = w_1$, and we take a lift to $\mathbb{Z}_N$ such that $\tilde{w}_1 = 0, 1 \bmod N$. Then $d\tilde{w}_1/2 = \tilde{w}_1^2 = 0, 1$. Then shifting the gauge field changes the flux quantization by

$$
\frac{N}{2\pi}\left(-\pi\frac{p}{N}d\tilde{w}_1 - 2\tilde{w}_1\left(-\pi\frac{p}{N}\tilde{w}_1\right)\right) = -p\left(\frac{d\tilde{w}_1}{2} - \tilde{w}_1^2\right) = 0 \bmod N\,.
\tag{48}
$$

Thus we find that changing CT to CT$U$ does not produce an non-trivial background $B$, and the symmetry fractionalization class remains unchanged.

Calculating $\nu_{\mathsf{CT}}$ for these matrices is straightforward. As this symmetry acts non-projectively on $\psi$, we just need to count signs under the transformation.

The case of $U \propto 1$ is trivial. Given that $\mathsf{CT}(\psi) = \psi$, the off-diagonal components of $\psi$ do not contribute, since they are complex and the real and imaginary parts transform with opposite signs. Only the Cartan subalgebra contributes: all diagonal components transform with sign $+1$, and therefore

$$\nu_{\mathsf{CT}} = N - 1. \tag{52}$$

Consider now $N$ even and $U \propto \mathrm{diag}(i\sigma_y, \ldots, i\sigma_y)$. We break $\psi$ into two-by-two blocks. As before, the off-diagonal blocks do not contribute, since real and imaginary parts transform with opposite signs. Let us, then, focus on a certain $2 \times 2$ block on the diagonal. Under $\mathsf{CT}(\psi) = U\psi U^{-1}$ such block transforms as

$$\begin{pmatrix} a & b \\ b^* & c \end{pmatrix} \mapsto (i\sigma_y)\begin{pmatrix} a & b \\ b^* & c \end{pmatrix}(-i\sigma_y) \equiv \begin{pmatrix} c & -b^* \\ -b & a \end{pmatrix}. \tag{53}$$

In other words, $a$ and $c$ are interchanged, and both $\mathrm{re}(b)$ and $\mathrm{im}(b)$ pick up a minus sign. This contributes $\nu = -2$. Adding up all diagonal blocks, and subtracting 1 for the trace, we arrive at

$$\nu'_{\mathsf{CT}} = -2(N/2) - 1 \equiv -N - 1. \tag{54}$$

In conclusion, the anomalies for $\mathsf{CT}$ in adjoint $SU(N)$ are $\nu_{\mathsf{CT}} = -1 \pm N$ when $N$ is even, and $\nu_{\mathsf{CT}} = -1 + N$ for odd $N$. The lower sign corresponds to a projective action on the Wilson lines.

For $N$ odd we find $\Delta\nu_{\mathsf{CT}} = 0$, while for $N$ even we find $\Delta\nu_{\mathsf{CT}} = 2N$, which equals 4 mod 8 when $N$ is 2 mod 4 and vanishes when $N$ is 0 mod 4.

## 3 QED and QCD in 1+1 dimensions

Consider a free $U(1)$ gauge field in $1+1d$, described by the Lagrangian

$$\mathcal{L} = \frac{1}{4e^2} da \wedge \star da + i\frac{\theta}{2\pi} da. \tag{55}$$

The parameter $\theta$ is as usual compact, $\theta \simeq \theta + 2\pi$.

The theory has a $U(1)$ one-form symmetry, the conserved current being the topological local operator $\star da$.[21] In addition, at $\theta = 0$ and at $\theta = \pi$ there is charge conjugation symmetry acting as $\mathsf{C}: a \to -a$. For the remaining of the discussion of this theory, we will add a massive charge 2 boson particle $B$ which transforms under charge conjugation as $\mathsf{C}: B \to B^*$. This reduces the one-form symmetry group to $\mathbb{Z}_2$. Standard arguments [63] show that there is a mixed anomaly between charge conjugation symmetry and the $\mathbb{Z}_2$ one-form symmetry at $\theta = \pi$ but not at $\theta = 0$. Since there is no mixed anomaly at $\theta = 0$, there will be no ambiguity in the 't Hooft anomaly of $\mathsf{C}$, as we will see.

An interesting question is to ask whether there is an 't Hooft anomaly for $\mathsf{C}$ (and not just a mixed anomaly with the one-form symmetry). Indeed, the anomalies of $\mathbb{Z}_2$ symmetries in bosonic systems are valued in $H^3(B\mathbb{Z}_2, U(1)) = \mathbb{Z}_2$. Therefore, one should be able to decide if $\mathsf{C}$ has an 't Hooft anomaly or not.

At $\theta = \pi$ this is a simple example of a situation where the anomaly depends on additional data, i.e. the choice of a fractionalization class.

---

[21]Topological local operators in $1+1d$ have been emphasized in [60]. See also [61] and the review [62] for the study of CFTs with topological local operators.

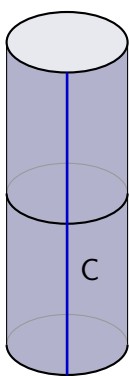

A standard way to diagnose 't Hooft anomalies in $1 + 1d$ is through the study of Hilbert spaces with topological defects.[22] According to [8, 64–66] (and see references therein), if the defect Hilbert space consists of Kramers doublets $\mathsf{T}^2 = -1$ then $\mathsf{C}$ has an anomaly. Otherwise, if the ground state is unique, then there is no anomaly for $\mathsf{C}$. As we will now see this prescription is insufficient in the theory (55).

Let us apply this criterion for the model (55). At infinite volume the model at $\theta = \pi$ has two vacua corresponding to the spontaneous breaking of $\mathsf{C}$ and at $\theta = 0$ the model has a unique vacuum. At $\theta = \pi$ the tension of the domain wall between the two vacua is infinite [66]. Now we take the theory on $S^1$ and place the $\mathsf{C}$ defect at some point along the $S^1$. From the discussion above we see that the ground state is unique at $\theta = 0$ while the Hilbert space is *empty* at $\theta = \pi$.[23]

We can therefore conclude that at $\theta = 0$ there is no anomaly for $\mathsf{C}$ (which could have been anticipated from the vacuum being trivially gapped at infinite volume at $\theta = 0$). For $\theta = \pi$ the question whether $\mathsf{C}$ has an 't Hooft anomaly requires further discussion since the Hilbert space is empty, as in the quantum mechanics examples in previous section.

We now construct two modifications of the theory at very high energies that lead to a trivial element of $H^3(B\mathbb{Z}_2, U(1))$ in the first case and non-trivial in the other case (i.e. no anomaly for charge conjugation in the first case and nonzero anomaly in the second case). In both of these modifications, one-form symmetry is completely broken by the heavy particles.

One obvious modification is to add a massive particle $\phi$ of charge 1 with

$$\mathsf{C}: \ \phi \to \phi^*. \tag{56}$$

We can add an arbitrary potential $V(|\phi|^2)$. It is clear that this model has no $\mathsf{C}$ anomaly since we could choose to condense $\phi$ – in that phase, there is a single trivial vacuum. One can also explicitly study the defect Hilbert space and see that there are no Kramers doublets and time-reversal symmetry in the defect Hilbert space satisfies $\mathsf{T}^2 = 1$.

Another possible modification of the theory at high energies is to add two species of a charge 1 boson, $\phi^1, \phi^2$, with a potential $V(|\phi^1|^2, |\phi^2|^2)$. The potential is constrained by a particular charge conjugation symmetry

$$\mathsf{C}: \ \phi^1 \to (\phi^2)^*, \quad \phi^2 \to -(\phi^1)^*. \tag{57}$$

---

[22]Anomalies also have implications for the Hilbert space at infinite volume without defects, since the defect cannot possibly be important at infinite volume.

[23]Another derivation of the same fact is that since $\star da$ is a topological local operator which is odd under $\mathsf{C}$, the only possible VEV is $\star da = 0$. On the other hand, the allowed VEVs in the theory with $\theta$ are $\star da = n - \theta/2\pi$ with intger $n$. Therefore, at $\theta = 0$ we have exactly one state in the defect Hilbert space while at $\theta = \pi$ there are no states. We thank N. Seiberg for providing this argument and for an illuminating discussion. Yet another derivation of the emptiness of the defect Hilbert space is that the space of end-points of $\mathsf{C}$ is empty, since lines that do not commute with topological local operators cannot end.

Note that $C^2 = -1$ on the scalar fields, but that is a gauge transformation and hence as a symmetry operator $C$ is of order 2. We can therefore say that $C$ is fractionalized on the gauge non-invariant fields $\phi^i$.

To understand the anomaly of $C$ in this situation, we can take several approaches. One is to again consider the defect Hilbert space. The defect Hilbert space is given by states which live on the double cover of the circle such that as we go from one copy to the next we implement a symmetry transformation. Therefore there is an approximate four-fold degeneracy whereby we can excite the particles $\phi^1, \phi^2$, and their complex conjugates. Each of these pairs is a Kramers doublet with $T^2 = -1$:

$$T|\phi^1\rangle = |\phi^2\rangle, \quad T|\phi^2\rangle = -|\phi^1\rangle, \quad T|(\phi^1)^*\rangle = |(\phi^2)^*\rangle, \quad T|(\phi^2)^*\rangle = -|(\phi^1)^*\rangle. \tag{58}$$

Upon taking into account non-perturbative corrections in the length of the circle the degeneracy between these two Kramers pairs is lifted and the ground state consists of a single Kramers pair. We conclude that the anomaly of the charge conjugation symmetry is now 1 mod 2. Another way to deduce the anomaly in the presence of the massive $\phi^1, \phi^2$ particles is to consider the following potential:

$$V = -M^2 \left(|\phi^1|^2 + |\phi^2|^2\right) + \lambda \left(|\phi^1|^2 + |\phi^2|^2\right)^2, \tag{59}$$

with $\lambda > 0$ and $M^2 > 0$. Minimizing the potential we see that the $U(1)$ gauge symmetry is everywhere Higgsed completely but there is a low energy mode which lives on a two-sphere. The low energy theory is $SU(2)_1$ WZW model and charge conjugation is embedded as the matrix $\mathrm{diag}(-1,-1,-1)$ acting on the embedding coordinates of the $S^2$. This symmetry is known to have a $\mathbb{Z}_2$ anomaly.

A possible interpretation of the results is to say that the massive particles transforming as (57) contribute 1 mod 2 to the $\mathbb{Z}_2$ anomaly.

A general lesson that will guide us later is that in (57), before gauging the $U(1)$ symmetry, the particles are not in a representation of $\mathbb{Z}_2$ – it is fractionalized to $\mathbb{Z}_4$.[24] This will be a general rule – massive particles can alter the anomaly only if before gauging they are in fractionalized representations of the zero-form symmetry $G$.

Above we have seen that the (non-gauge invariant fields) $\phi^i$ are in a projective representation of $C$. It is important to emphasize that this is independent of whether we use the above action of $C$ or compose $C$ with an arbitrary gauge transformation which implements a rotation by angle $\alpha$, $U_\alpha$. Indeed, $CU_\alpha$ acts by

$$CU_\alpha: \quad \phi^1 \to e^{i\alpha}\left(\phi^2\right)^*, \quad \phi^2 \to -e^{i\alpha}\left(\phi^1\right)^*. \tag{60}$$

And it is straightforward to verify that on the fields $\phi^i$ we have $(CU_\alpha)^2 = -1$. Therefore it impossible to use gauge transformations to make the particles $\phi^i$ sit in a non-fractionalized representation of $C$. This will be important later.

Finally, we would like to give a simple interpretation of how it is possible for massive particles to contribute to the $\mathbb{Z}_2$ anomaly in theories with one-form symmetry. Consider the theory of the single massive complex particle $\phi$ and couple to the $U(1)$ and $C$ background gauge fields. This leads to the gauge group $O(2) = SO(2) \rtimes \mathbb{Z}_2 \cong Pin^+(2)$. On the other hand, for the case of the two particles $\phi^{1,2}$ with the action of $C$ in (57) the gauge group is[25]

$$\frac{SO(2) \rtimes \mathbb{Z}_4}{\mathbb{Z}_2} \cong Pin^-(2). \tag{61}$$

---

[24]Since the one-form symmetry is $\mathbb{Z}_2$, the map $\rho$ (2) here is trivial. The fractionalization classes are $H^2_\rho(\mathbb{Z}_2, \mathbb{Z}_2) = \mathbb{Z}_2$.

[25]The two fractionalization classes considered here correspond to the two extensions $Pin^\pm(2)$ of $\mathbb{Z}_2$ charge conjugation by $U(1) = SO(2) \cong Spin(2)$. In particular, for these fractionalization classes, the charge conjugation symmetry does not participate in a two-group.

In the case of $O(2) = SO(2) \rtimes \mathbb{Z}_2 \cong Pin^+(2)$, the $O(2)$ bundles always have integer magnetic flux. In the case of $\frac{SO(2) \rtimes \mathbb{Z}_4}{\mathbb{Z}_2}$, we can have half-integer magnetic fluxes for $SO(2)$, if the $\mathbb{Z}_2$ bundle for the charge conjugation symmetry cannot be lifted to a $\mathbb{Z}_4$ bundle. The appearance of half-integer $SO(2)$ fluxes in the presence of non-trivial background fields for C is why the massive particles can affect the anomalies. Since the half-integer flux modifies the $\theta$ periodicity, $\theta = \pi$ no longer preserves the charge conjugation symmetry, and this contributes an 't Hooft anomaly for the charge conjugation symmetry. A similar interpretation is possible in all the examples below – fractionalization classes are essentially a way to allow for more general gauge bundles than naively appears possible.

The statements above about the anomaly of the $\mathbb{Z}_2$ symmetry in free $U(1)$ gauge theory can be understood intrinsically from the point of view of the low-energy $\mathbb{Z}_2$ gauge theory:

$$S = \pi \int a_{(0)} \cup \delta b_{(1)}, \tag{62}$$

where $a_{(0)}, b_{(1)}$ are dynamical $\mathbb{Z}_2$ 0- and 1-cochains, respectively, and $\delta$ is the coboundary operator. The content of this topological theory is that it has two vacua and a topological $\mathbb{Z}_2$ symmetry line that is a domain wall between these two vacua.

We couple this system to a background $\mathbb{Z}_2$ gauge field $A_{(1)}$ for the $\mathbb{Z}_2$ zero-form symmetry. $A_{(1)}$ is represented by a closed one-cochain.

$$S = \pi \int \left( a_{(0)} \cup \delta b_{(1)} + A_{(1)} \cup b_{(1)} \right). \tag{63}$$

The gauge transformations act as ($\lambda$ is the parameter for the background gauge transformation of the $\mathbb{Z}_2$ symmetry)

$$a_{(0)} \to a_{(0)} + \lambda, \quad b_{(1)} \to b_{(1)} + \delta \omega, \quad A_{(1)} \to A_{(1)} + \delta \lambda. \tag{64}$$

The action is perfectly gauge invariant and there is no anomaly for the $\mathbb{Z}_2$ symmetry. We can see that the $\omega$ gauge transformations leave the action invariant by virtue of $A_{(1)}$ being flat.

An interesting coupling that we could add is

$$S = \pi \int \left( a_{(0)} \cup \delta b_{(1)} + A_{(1)} \cup b_{(1)} + a_{(0)} \cup A_{(1)} \cup A_{(1)} \right). \tag{65}$$

The background gauge transformation now is

$$A_{(1)} \to A_{(1)} + \delta \lambda, \quad a_{(0)} \to a_{(0)} + \lambda, \quad b_{(1)} \to b_{(1)} + \lambda \cup A_{(1)} + A_{(1)} \cup \lambda + \lambda \cup \delta \lambda. \tag{66}$$

The action is not invariant, but transforms by an anomalous shift. To cancel the shift we can introduce a bulk term $\pi \int_{3d} A_{(1)} \cup A_{(1)} \cup A_{(1)}$. To see that the bulk is sufficient to make the theory invariant under the background gauge transformation of $A_{(1)}$, let us write the boundary-bulk terms together as

$$\pi \int_{3d} \left( \delta a_{(0)} + A_{(1)} \right) \cup \left( \delta b_{(1)} + A_{(1)} \cup A_{(1)} \right) \tag{67}$$

$$= \pi \int_{2d} \left( a_{(0)} \cup \delta b_{(1)} + A_{(1)} \cup b_{(1)} + a_{(0)} \cup A_{(1)} \cup A_{(1)} \right) + \pi \int_{3d} A_{(1)} \cup A_{(1)} \cup A_{(1)} \bmod 2\pi,$$

where the equality used $\delta A_{(1)} = 0 \bmod 2$. The left hand side is invariant under background gauge transformation, and thus the right hand side, which consists of (65) and the bulk term $\pi \int_{3d} A_{(1)} \cup A_{(1)} \cup A_{(1)}$, is also invariant under background gauge transformation.

We therefore see that in some version of the $\mathbb{Z}_2$ gauge theory in $1+1d$ there is an anomaly for the zero-form $\mathbb{Z}_2$ symmetry and in another version there is no anomaly. The theories only differ by the couplings to background fields.

## 3.1 Fermionic $\mathbb{Z}_2$ symmetry in 1+1 dimensions

In this subsection we repeat the discussion above in a fermionic theory with $\mathbb{Z}_2$ unitary symmetry. In theories with fermions a $\mathbb{Z}_2$ symmetry with algebra $g^2 = 1$ has anomalies classified by $\Omega_3^{Spin}(B\mathbb{Z}_2) = \mathbb{Z}_8$. We will study a system with a $\mathbb{Z}_2$ one-form symmetry and a mixed anomaly between the one-form and zero-form symmetry. As a result, the ordinary familiar zero-form anomaly in $\mathbb{Z}_8$ is ambiguous in units of 4 mod 8 depending on the fractionalization class.

Consider QED$_2$ with a massless charge 2 Dirac fermion $\Psi$ and a charge 2 boson $\Phi$. We will denote by $\Psi_L$ and $\Psi_R$ the respective left and right moving complex fermions. This theory has an axial $\mathbb{Z}_2$ symmetry:

$$g: \quad \Psi_L \to -\Psi_L\,, \quad \Psi_R \to \Psi_R\,, \quad \Phi \to \Phi\,. \tag{68}$$

We can ask if $g$ has an 't Hooft anomaly.

One approach is to condense $\Phi$ – then the low energy theory consists of one massless Dirac fermion and the $\mathbb{Z}_2$ gauge theory studied in the previous subsection. The Dirac fermion contributes 2 mod 8 to the anomaly, but the $\mathbb{Z}_2$ gauge theory, as we saw, may either contribute the term $\pi \int_{3d} A_{(1)} \cup A_{(1)} \cup A_{(1)}$ to the inflow invertible phase or not. The term $\pi \int_{3d} A_{(1)} \cup A_{(1)} \cup A_{(1)}$ amounts to a contribution of 4 mod 8 to the anomaly. Therefore we infer that the anomaly for $\mathbb{Z}_2$ is either 2 mod 8 or $(-2)$ mod 8, depending on the fractionalization class. Indeed by a gauge transformation we can write an equivalent expression for the action of $g$:

$$g e^{-i\pi/2}: \quad \Psi_L \to \Psi_L\,, \quad \Psi_R \to -\Psi_R\,, \quad \Phi \to -\Phi\,, \tag{69}$$

where $e^{-i\pi/2}$ stands for a 90 degrees gauge transformation. Since the boson $\Phi$ is not fractionalized and it can be easily gapped it does not contribute to the anomaly, so, in this presentation, the anomaly $-2$ mod 8 is more natural.

Of course $g$ and $g e^{-i\pi/2}$ define the exact same symmetry (only differing by a gauge transformation). This trick of composing $g$ with a gauge transformation allows to quickly predict that the two possible outcomes for the anomaly are $\pm 2$ mod 8. This is a general theme, which we have already encountered in the previous section.

To see more formally why this heuristic rule works we can add a massive charge 1 boson $u$ and break the one-form symmetry completely. We must specify how the new boson $u$ transforms under $g$, and we have two choices, which we denote by $g$, and $g'$, which are now genuinely different symmetries, not related to each other by gauge transformations:

$$\begin{aligned} g: \quad & \Psi_L \to -\Psi_L\,, \quad \Psi_R \to \Psi_R\,, \quad \Phi \to \Phi\,, \quad u \to u\,, \\ g': \quad & \Psi_L \to -\Psi_L\,, \quad \Psi_R \to \Psi_R\,, \quad \Phi \to \Phi\,, \quad u \to iu\,. \end{aligned} \tag{70}$$

Note that both $g^2 = 1$ and $g'^2 = 1$. Since now the one-form symmetry is completely broken the anomaly of both $g$ and $g'$ should be given by a fixed, unambiguous element of $\mathbb{Z}_8$. In the case of $g$ the boson $u$ is in a non-fractionalized representation and hence the anomaly is given by the low energy fields and is equal to 2 mod 8. In the case of $g'$ the massive boson $u$ is in a fractionalized representation and to account for its contribution to the anomaly it is again useful to perform a $e^{-i\pi/2}$ gauge transformation and obtain

$$g' e^{-i\pi/2}: \quad \Psi_L \to \Psi_L\,, \quad \Psi_R \to -\Psi_R\,, \quad \Phi \to -\Phi\,, \quad u \to u\,. \tag{71}$$

Now the boson $u$ is in a non-fractionalized representation and does not contribute to the anomaly, but the price to pay is that the right moving fermion now picks a minus sign and hence the anomaly is $-2$ mod 8.[26]

---

[26]The same conclusion can be arrived at by observing that the gauge invariant operators are $\bar{\Psi}_L u^2, \bar{\Psi}_R u^2$, and in the case of the symmetry $g$ it is the left moving composite fermion that picks up a minus sign while for $g'$ it is the right moving composite fermion that picks a sign. These composites directly translate to the massless modes if $u$ condenses.

In summary, we see that in fermionic theories in $1 + 1d$ with $G = \Gamma = \mathbb{Z}_2$ and with a mixed anomaly between the zero-form and one-form symmetries, the anomaly of $G$ is classified as usual by $\mathbb{Z}_8$, but it depends on a choice of a fractionalization class in $H^2(\mathbb{Z}_2, \mathbb{Z}_2) = \mathbb{Z}_2$ (in this case $\rho$ is trivial).

### 3.1.1 The mixed $\mathbb{Z}_2 - (-1)^F$ anomaly

Here we are dealing with a fermionic theory, and therefore we have to discuss the symmetry $(-1)^F$ as well

$$(-1)^F : \Psi_L \to -\Psi_L, \quad \Psi_R \to -\Psi_R, \quad \Phi \to \Phi. \tag{72}$$

Mixed $\mathbb{Z}_2 - (-1)^F$ anomalies in $1 + 1d$ are classified by $\mathbb{Z}_2$, namely the anomaly is either 0 or 1 mod 2.

QED$_2$ with a massless charge 2 Dirac fermion $\Psi$ and a charge 2 boson $\Phi$ has a vanishing mixed anomaly between $g$ (68) and $(-1)^F$ since there are two left moving Majorana fermions transforming under both $(-1)^F$ and $g$. We need to discuss if this mixed anomaly can be altered by changing fractionalization classes. So far we discussed the fractionalization class corresponding to $B = \frac{1}{2} q_\rho(A)$, where we denote the spin structure by $\rho$, and the $g$ background gauge field by $A$. We have seen that this allows to shift the pure $\mathbb{Z}_2$ anomaly in units of 4 mod 8 but it does not affect the mixed $\mathbb{Z}_2 - (-1)^F$ anomaly.

Indeed, to change the mixed anomaly we would have to consider the fractionalization class $B = \text{Arf}$, which, roughly speaking, fractionalizes fermion number symmetry. However, there is no sense in adding relativistic massive degrees of freedom with such a fractionalization class since it requires an odd number of Majorana fermions. In addition, a $0 + 1$ dimensional model with an odd number of Majorana fermions breaks fermion number symmetry due to a nonzero expectation value of a product of an odd number of fermions.

Therefore, the mixed anomaly between $(-1)^F$ and $\mathbb{Z}_2$ symmetry is unambiguously 0 mod 2. In the following subsection we will see a model where this mixed anomaly is 1 mod 2, but again, it will be unambiguous.

## 3.2 A few remarks on adjoint QCD in 1+1 dimensions

We consider $SU(N)$ gauge theory with a massless non-chiral Majorana fermion in the adjoint representation

$$S = \int d^2x \left( \frac{-1}{4g_{YM}^2} \text{tr} \, F^2 + i \, \text{tr} \, \Psi^T \slashed{D} \Psi \right). \tag{73}$$

The study of the infrared dynamics of $1 + 1d$ adjoint gauge theory goes back to [67, 68] and there was progress on it recently, see [69–75] and references therein.

This theory has a $\mathbb{Z}_2^A$ axial symmetry acting as $\Psi_L \to -\Psi_L, \Psi_R \to \Psi_R$. Furthermore, the theory has $\mathbb{Z}_N$ one-form symmetry and there is a mixed anomaly between the zero-form $\mathbb{Z}_2$ symmetry and the one-form $\mathbb{Z}_N$ symmetry for even $N$ [69]. Now we will attempt to compute the mod 8 anomaly for $\mathbb{Z}_2^A$ – which we anticipate is ambiguous for even $N$ but unambiguous for odd $N$ (ambiguous in the sense that it depends on the choice of action of $\mathbb{Z}_2^A$ on lines). In addition, for even $N$, the model has a mixed anomaly between the $\mathbb{Z}_2^A$ axial symmetry and fermion number symmetry, which as we have discussed above, will be unambiguously 1 mod 2.

First we will consider the anomaly as computed in the ultraviolet taking into account the possible dependence on the choice of fractionalization classes, and then we will contrast our results with some conjectures about the infrared dynamics of this theory.

We can compute the anomaly in the UV from the fact that $N^2 - 1$ chiral Majorana fermions flip sign under $\mathbb{Z}_2^A$. Therefore we find $N^2 - 1$ mod 8. For $N$ even the anomaly is either 3 or 7 mod 8 depending on whether $N = 2$ mod 4 or $N = 0$ mod 4, respectively and for $N$ odd

the anomaly always vanishes. We can compose $\mathbb{Z}_2^A$ with a gauge transformation in order to scan over the possible fractionalization classes. It is sufficient to consider diagonal matrices for the gauge transformations $U = \alpha\,\mathrm{diag}(-1,\ldots,-1,1,\ldots,1)$ with the minus sign appearing $p$ times on the diagonal and $\alpha^N = (-1)^p$. If we consider $\mathbb{Z}_2^A \cdot U$ as our new axial symmetry then $2p(N-p)$ right moving fermions pick up a minus sign under $\mathbb{Z}_2^A \cdot U$ while $N^2 - 1 - 2p(N-p)$ left moving fermions pick up a minus sign. The anomaly is thus $N^2 - 1 - 4p(N-p)$. We see that the correction $4p(N-p)$ is divisible by 4 signifying that the anomaly can only jump by 4 mod 8 due to changing the fractionalization class. For even $N$, by choosing any odd $p$ we therefore can shift the anomaly of $\mathbb{Z}_2^A$ from $N^2 - 1$ mod 8 to $N^2 + 3$ mod 8. For odd $N$, $4p(N-p)$ is always divisible by 8 and the anomaly is independent of the fractionalization class, in line with our expectations.

In summary, $\mathbb{Z}_2^A$ has anomaly 3 or 7 mod 8 for even $N$ and no anomaly for odd $N$. Further, for even $N$ we have a mixed anomaly with $(-1)^F$ and no mixed anomaly for odd $N$.[27] It is now time to see how this is matched by the conjectured infrared phase of this theory.

To simplify the infrared dynamics we consider the $\mathbb{Z}_2^A$ invariant four-fermion deformation of [69, 70] (for a similar discussion in $\mathrm{QED}_2$ see [76])

$$\mathrm{tr}(\Psi_L \Psi_R)\,\mathrm{tr}(\Psi_L \Psi_R)\,. \tag{74}$$

This deformation preserves the one-form symmetry, fermion number symmetry and it also preserves the axial symmetry. The ground state is conjectured to be doubly degenerate due to a fermion condensate that breaks the $\mathbb{Z}_2^A$ symmetry spontaneously.

For even $N$ all the Wilson lines in a representation with the number of boxes not divisible by $N/2$ obey an area law. The Wilson lines with the number of boxes divisible by $N/2$ obey instead a perimeter law [69, 70]. For odd $N$, all the Wilson lines obey an area law.

The low-energy theory is therefore again $\mathbb{Z}_2$ gauge theory for both even and odd $N$. The topological line operator for even $N$ originates in the ultraviolet from a Wilson line in a representation with $N/2$ boxes. For odd $N$ the topological line originates from a finite tension domain wall (kink).

For odd $N$ the one-form symmetry does not couple to the infrared degrees of freedom of $\mathbb{Z}_2$ gauge theory. This is why the dependence on the fractionalization class completely drops out and the anomaly in this case is universally 0 mod 8.

For even $N$ we now have to understand how the $\mathbb{Z}_2$ gauge theory reproduces the various values of the anomaly we have seen above. As we have shown, the 't Hooft anomaly for the zero-form symmetry in ordinary, bosonic, $\mathbb{Z}_2$ gauge theory is either 0 or 4 mod 8. However, here we are dealing with a fermionic theory, which allows an additional term

$$\pi \int_{2d} a_0 \cup \mathrm{Arf}\,, \tag{75}$$

with the Arf invariant of the Poincaré dual of $a_0$. The Lagrangian (75) makes sense for constant $a_0$. A precise Lagrangian and treatment of the anomalies of this theory is in appendix D. Here we just summarize the main results.

The two vacua of $\mathbb{Z}_2$ gauge theory now have different Arf counter-terms (i.e. carry different invertible phases) and therefore the defect Hilbert space would contain an extra Majorana fermion [10, 14]. Furthermore the spin structure entering in the Arf invariant can be shifted by the background gauge field of $\mathbb{Z}_2^A$ which allows to change the chirality under time-reversal of this extra Majorana fermion in the defect Hilbert space. Such a modification of $\mathbb{Z}_2$ gauge theory allows to obtain all the required values for the anomaly.

---

[27]A similar analysis shows that the anomalies of $\mathbb{Z}_2^A(-1)^F$ (the diagonal $\mathbb{Z}_2$ subgroup of $\mathbb{Z}_2^A$ and $(-1)^F$), are 1 or 5 mod 8, depending on the fractionalization class for even $N$, while for odd $N$ they are again 0 mod 8.

It remains to explain why the ultraviolet gauge theory dictates the appearance of the term (75). We have a condensate $\langle \Psi_L \Psi_R \rangle \neq 0$ breaking spontaneously the $\mathbb{Z}_2^A$ symmetry. Due to the mixed anomaly between $(-1)^F$ and $\mathbb{Z}_2^A$ symmetry, the two vacua must have a different Arf SPT phase. The term (75) therefore reproduces this mixed anomaly. It is very nice that this modification of the $\mathbb{Z}_2$ gauge theory also simultaneously reproduces the $\mathbb{Z}_2^A$ anomalies of the theory.

# 4 Time-reversal and symmetry fractionalization in 2+1 dimensions

We study $2+1d$ systems with time-reversal symmetry. There are two versions of the time-reversal symmetry operator, one of which we denote $\mathsf{T}$ and the other we denote by $\mathsf{CT}$. If charge conjugation $\mathsf{C}$ is a symmetry in its own right then both $\mathsf{T}$ and $\mathsf{CT}$ exist simultaneously. The algebra of these symmetries is

$$\mathsf{T}^2 = (-1)^F, \qquad (\mathsf{CT})^2 = (-1)^F. \tag{76}$$

Charge conjugation $\mathsf{C}$ is a symmetry in gauge theory if the gauge group admits an outer automorphism and the matter fields respect the symmetry, which will be the case in examples below. We denote the associated symmetry groups by $G = \mathbb{Z}_4^{\mathsf{T}}$ and $G = \mathbb{Z}_4^{\mathsf{CT}}$ respectively. The anomaly classification for these time-reversal symmetries is well known [10, 77–80], they take values in $\Omega_{\mathrm{Pin}^+}^4(\mathrm{pt}) = \mathbb{Z}_{16}$, and we denote them by $\nu_{\mathsf{T}}$ and $\nu_{\mathsf{CT}}$.

Suppose the system also has $\Gamma = \mathbb{Z}_n$ one-form symmetry in addition to the time-reversal symmetry. Let us define time-reversal $\mathsf{T}$ and $\mathsf{CT}$ to be such that $\mathsf{CT}$ leaves the one-form symmetry charges invariant, while $\mathsf{T}$ acts on the one-form symmetry by the $\mathbb{Z}_2$ charge conjugation symmetry on the $\mathbb{Z}_n$ charges: $q \to -q \bmod n$.[28]

One-form symmetries can have an 't Hooft anomaly in $2+1d$. With no other symmetries, in theories with fermions, the anomaly is valued is $p \in \mathbb{Z}_n$ and the anomaly is determined from the spin of the generator, $h = p/2n$ [81]. In time-reversal invariant theories, the allowed anomalies for one-form symmetry are restricted. The anomaly for the one-form symmetry is now classified by the subgroup of $\Omega_{\mathrm{Pin}^+}^4(B^2\mathbb{Z}_n)$ generated by the elements that are not in $\Omega_{\mathrm{Pin}^+}^4(pt)$. The anomaly is in $\mathbb{Z}_4$ for $n$ even and trivial for $n$ odd. This is true for both $\mathsf{T}$ and $\mathsf{CT}$.

To understand the origin of this $\mathbb{Z}_4$, we observe that in time-reversal invariant theories, the generator of the $\mathbb{Z}_n$ one-form symmetry has to be left (almost) invariant under time-reversal; the generating line $a$ of the symmetry therefore satisfies either

- $\mathsf{T}a = a$, in such case its spin $h$ satisfies $h = -h \bmod 1$

- $\mathsf{T}a = a\psi$ with transparent fermion line $\psi$, in such case its spin $h$ satisfies $h + \frac{1}{2} = -h \bmod 1$.

Thus the generator of the $\mathbb{Z}_n$ one-form symmetry can only have spin $h = 0, \frac{1}{2}, \pm\frac{1}{4} \bmod 1$. The same discussion holds for $\mathsf{CT}$.

One can arrive at the same conclusion about the allowed anomalies in time-reversal invariant theories by investigating the anomaly inflow term $2\pi i \frac{p}{2n} \int_{4d} \mathcal{P}(B)$. On spin orientable manifolds $p = 0, ..., n-1$ describes the distinct allowed possibilities [26, 81]. For this counter-term to be

---

[28]Note an important fact: $\mathsf{T}$ reverses the electric charge and hence the electric field is odd under time-reversal while the magnetic field is even. Since the electric two-form gauge field $B$ couples to the electromagnetic field via $B^{\mu\nu}F_{\mu\nu}$, as a two-form, $B$ is even under $\mathsf{T}$ and odd under $\mathsf{CT}$ while the corresponding one-form symmetry charge, $\int_\Sigma \star F$, is odd under $\mathsf{T}$ and even under $\mathsf{CT}$. This reversal of odd/even between the charges and the corresponding background fields is particular to time-reversal symmetry.

time-reversal invariant we must require $2p = 0$ mod $n$. For even $n$ the solutions are $p = n/2$ mod $n$ or $p = 0$ mod $n$ which allow for the spins $h = 0, \frac{1}{2}, \pm\frac{1}{4}$ while for odd $n$ only $p = 0$ mod $n$ is allowed but that corresponds to a trivial invertible theory (which furthermore remains trivial on non-orientable manifolds, unlike the even $n$ case).

The one-form anomaly in time-reversal invariant theories can be written using the $3 + 1d$ bulk action

$$2\pi h \int q_\rho (B \bmod 2), \quad h \in \left\{ 0, \frac{1}{2}, \pm\frac{1}{4} \right\}, \tag{77}$$

where $B$ is the background for $\mathbb{Z}_n$ one-form symmetry for even $n$, $q$ is the $\mathbb{Z}_4$-valued quadratic function that depends on Wu$_3$ structure $\rho$ [23, 82, 83], and on $pin^+$ manifolds $\rho = w_1 \cup \rho_1$ with $\rho_1$ being the $pin^+$ structure. For $n = 2$, the effective action is discussed in [84].

The anomaly (77) for $h = \frac{1}{2}$ mod 1 can be interpreted as a mixed anomaly between the one-form symmetry and time-reversal symmetry. Similar mixed anomalies in $2 + 1d$ were discussed in [85]. Indeed, as remarked above, $h = 1/2$ is trivial without time-reversal symmetry because we can simply dress the generator $a$ with the transparent fermion and gauge this spin-less line. But with time-reversal symmetry $a\psi$ would be a Kramers doublet:

$$h = \frac{1}{2}: \quad \pi \int q_\rho (B \bmod 2) = \pi \int B \cup B = \pi \int B \cup (w_2 + w_1^2) = \pi \int B \cup w_1^2 \mod 2\pi, \tag{78}$$

where in the second equality we used the Wu formula $x_2 \cup x_2 = x_2 \cup (w_2 + w_1^2)$ for $\mathbb{Z}_2$ two-cocycle $x_2$ on 4-manifolds [86], and in the last equality we used $w_2$ is trivial on $pin^+$ manifolds [87].

## 4.1 Different symmetry fractionalizations and their anomalies

Due to the one-form symmetry, the anomalies for the time-reversal symmetry are potentially ambiguous. This can be traced to the fact that the coupling to an unorientable manifold is ambiguous due to the existence of fractionalization classes. The story is pretty similar both for T and CT even though the former does not act on the two-form background gauge field for the one-form symmetry while the latter flips the sign of the two-form background gauge field. We will only discuss explicitly the case of T and quote the results for CT at the end.

The corresponding fractionalization classes are labeled by $\mathbb{Z}_2$. The non-trivial fractionalization class corresponds to the shift $B \to B + dw_1/2$, where $dw_1/2$ mod $n$ is the image of $w_1$ under the Bockstein for the short exact sequence $1 \to \mathbb{Z}_n \to \mathbb{Z}_{2n} \to \mathbb{Z}_2 \to 1$.

If the one form symmetry has 't Hooft anomalies, the shift $B \to B + dw_1/2$ would lead to various "jumps" for the time-reversal symmetry anomaly.

- When $h = 0$: the one-form symmetry does not have anomaly, and the shift does not produce additional anomaly.

- When $h = \frac{1}{2}$, the one-form symmetry has a mixed anomaly with the time-reversal symmetry, and the shift produces additional time-reversal anomaly $\nu_T = 8$ as described by the bulk effective action

$$\pi \int w_1^2 \cup w_1^2 = \pi \int w_1^4, \tag{79}$$

  where we used $dw_1/2 \mod 2 = w_1^2$.

- When $h = \pm\frac{1}{4}$, using the quadratic property $q(x + y) = q(x) + q(y) + 2x \cup y$ mod 4 for $x = B, y = w_1^2$, one finds the shift produces additional time-reversal anomaly $\nu_T = \pm 4$, in addition to extra mixed anomaly between the one-form symmetry and time-reversal symmetry, as described by the bulk effective action

$$\pi \int B \cup w_1^2 \pm \frac{\pi}{2} \int q_\rho(w_1^2). \tag{80}$$

Let us explain why the last term in (80) is the bulk effective action for $v_\mathsf{T} = \pm 4$. We can describe the invertible phase with $v_\mathsf{T} = \pm 2$ using an auxiliary dynamical $\mathbb{Z}_2$ two-form gauge field $b$, with the action [23]

$$\frac{\pi}{2} \int q_\rho(b). \tag{81}$$

The signs of $v_\mathsf{T} = \pm 2$ are related by shifting the $pin^+$ structure $\rho_1$ in $\rho = w_1 \cup \rho_1$ by $\rho_1 \to \rho_1 + w_1$. For a suitable convention of $\rho_1$, we can take the sign to be $v_\mathsf{T} = +2$ for the above theory. Under the field redefinition $b \to b + w_1^2$, using the quadratic function property $q(x + y) = q(x) + q(y) + 2x \cup y \mod 4$, the action changes to

$$\frac{\pi}{2} \int q_\rho(b) + \frac{\pi}{2} \int q_\rho(w_1^2) + \pi \int b \cup w_1^2 = -\frac{\pi}{2} \int q_\rho(b) + \frac{\pi}{2} \int q_\rho(w_1^2) \mod 2\pi, \tag{82}$$

where the equality uses $[w_2] = 0$ on $pin^+$ manifold, and thus $w_1^2 = w_2 + w_1^2 = v_2$, $w_1^2 \cup b = v_2 \cup b = b \cup b$. Since the field redefinition does not change the partition function,

$$Z_{v_\mathsf{T} = \pm 2} = (Z_{v_\mathsf{T} = \pm 2})^* e^{\frac{\pi i}{2} \int q_\rho(w_1^2)}, \tag{83}$$

where $Z_{v_\mathsf{T} = \pm 2}$ is the partition function for the two-form gauge theory (81) that is the invertible theory with $v_\mathsf{T} = \pm 2$. Thus we find that the theory with effective action $\pm \frac{\pi}{2} \int q_\rho(w_1^2)$ corresponds to the anomaly

$$v_\mathsf{T} = (\pm 2) - (\mp 2) = \pm 4. \tag{84}$$

This concludes our discussion of the time-reversal anomaly. We see that the jumps in the anomaly due to the different fractionalization classes are always in multiples of 4 mod 16.

We summarize the results in the following table:

| | $\mathbb{Z}_n^{(1)}$ | $\mathbb{Z}_n^{(1)} - \mathbb{Z}_4^\mathsf{T}$ | $\Delta v_\mathsf{T} \mod 16$ |
|---|---|---|---|
| $h \in \mathbb{Z}$ | $X$ | $X$ | 0 |
| $h \in 1/2 + \mathbb{Z}$ | $X$ | $\checkmark$ | 8 |
| $h \in \pm 1/4 + \mathbb{Z}$ | $\checkmark$ | $X$ | $\pm 4$ |

$$(85)$$

The column $\mathbb{Z}_n^{(1)}$ stands for pure one-form symmetry anomaly, the column $\mathbb{Z}_n^{(1)} - \mathbb{Z}_4^\mathsf{T}$ stands for the mixed anomaly between time-reversal symmetry and one-form symmetry and, finally, the column with $\Delta v_\mathsf{T} \mod 16$ describes the allowed jumps due to different fractionalization classes.

We now proceed to study symmetry fractionalization in $QCD_3$ theories, where we will apply the results above to test some proposals about the behavior of such theories in the deep infrared.

## 4.2 $SU(N)$ with adjoint quark

Let us remind some basic facts about adjoint QCD in $2 + 1$ dimensions, namely $SU(N)$ gauge theory with Chern-Simons level $k$ and an adjoint Majorana fermion $\lambda$ valued in the $\mathfrak{su}(N)$ Lie algebra. For recent work on QCD theories in 2+1 dimensions of the type discussed here see [31, 88–94]. One has to specify the Chern-Simons level $k$ for the gauge group and one can write a mass term $m \operatorname{tr} \lambda\lambda$. An important consistency condition for this theory to exist is

$$k + N/2 \in \mathbb{Z}. \tag{86}$$

All these theories have a one-form symmetry $\mathbb{Z}_N$ with anomaly $p = k + N/2 \mod N$. The case particularly interesting to us here is that with even $N$ and $k = 0$. In this special case the theory

with a massless adjoint quark, $m = 0$, enjoys time-reversal symmetry. Indeed, the one form symmetry anomaly is $p = N/2$ which leads to the spin of the generator being $h = \pm\frac{1}{4}$, which is one of the values allowed in time-reversal invariant theories, as we have seen above.

A free massless Majorana fermion in $3d$ has a $\mathbb{Z}_4^{\mathsf{T}}$ time-reversal symmetry, which can be taken to act as

$$\mathsf{T} : \lambda \to \gamma^0 \lambda \,. \tag{87}$$

Since $\mathsf{T}$ is anti-unitary, it acts on a Dirac fermion $\psi = \lambda_1 + i\lambda_2$ as

$$\mathsf{T} : \psi \to \gamma^0 \psi^* \,. \tag{88}$$

On a Dirac fermion we can also define the action of $\mathbb{Z}_4^{\mathsf{CT}}$, which acts instead as

$$\mathsf{CT} : \psi \to \gamma^0 \psi \,. \tag{89}$$

For even $N$ and a massless adjoint fermion, $3d$ $SU(N)$ adjoint QCD with $k = 0$ is $\mathbb{Z}_4^{\mathsf{T}}$ and $\mathbb{Z}_4^{\mathsf{CT}}$ symmetric.[29] Let us discuss the 't Hooft anomalies for $\mathbb{Z}_4^{\mathsf{T}}$ and $\mathbb{Z}_4^{\mathsf{CT}}$ and how they are matched in the infrared.

Let us first turn to the $\mathbb{Z}_4^{\mathsf{T}}$ time-reversal anomaly, which acts on the hermitian fermions as

$$\mathsf{T} : \lambda_{ij} \to \gamma^0 \lambda_{ji} \,, \tag{92}$$

and we can thus instantly infer the time-reversal anomaly since each of the Majorana fermions contributes $+1$:

$$\nu_{\mathsf{T}} = N^2 - 1 \mod 16 = \begin{cases} -1, & N = 0 \mod 4, \\ 3, & N = 2 \mod 4. \end{cases} \tag{93}$$

We already know that these results should be taken with a grain of salt since the theory has a one-form symmetry and the spin of the generator is $\pm\frac{1}{4}$ so the anomaly for time-reversal symmetry depends on additional data, namely, the fractionalization classes. To probe the fractionalization classes we can combine $\mathsf{T}$ with $U \in SU(N)$ in (92) such that $\mathsf{T}U$ preserves the reality of the fermion and $(\mathsf{T}U)^2 = (-1)^F$. This can be accomplished with

$$U = \alpha \operatorname{diag}\left(-1_p, 1_{N-p}\right), \tag{94}$$

for some $p \in [0, N]$ and $\alpha$ is a phase obeying $\alpha^N = (-1)^p$. The anomaly associated to $\mathsf{T}U$ is (30), namely $\nu_{\mathsf{T}U} = (N - 2p)^2 - 1$. From this we find that the anomaly $\nu_{\mathsf{T}}$ shifts under $U$ as

$$\nu_{\mathsf{T}} - \nu_{\mathsf{T}U} = 4p(N - p) = \begin{cases} 0 & \mod 16, \ p \text{ even}, \\ -4 & \mod 16, \ N = 0 \mod 4, \ p \text{ odd}, \\ 4 & \mod 16, \ N = 2 \mod 4, \ p \text{ odd}. \end{cases} \tag{95}$$

Therefore, the time-reversal anomaly for $\mathsf{T}$ and $\mathsf{T}U$ can be either 3 or $-1$ for any even $N$, depending on the fractionalization class.

---

[29] The time-reversal symmetry $\mathsf{T}$ commutes with the gauge group, since it is anti-unitary and the electric charge is odd, and thus

$$\mathsf{T} e^{i\alpha Q} = e^{i\alpha Q} \mathsf{T} \,. \tag{90}$$

for any gauge parameter $\alpha$ and where $Q$ is the charge operator. If the gauge group is $U(1)$, this is often referred to as $U(1) \times \mathbb{Z}_4^{\mathsf{T}}$ in condensed matter literature. The time-reversal symmetry $\mathsf{CT}$ does not commute with the gauge group, since it is anti-unitary and the electric charge is even, and thus for $N > 2$, (for $N = 2$ the sign does not matter, and it coincides with the case of $\mathsf{T}$)

$$\mathsf{CT} e^{i\alpha Q} = e^{-i\alpha Q} \mathsf{CT} \,. \tag{91}$$

When the gauge group is $U(1)$, this is often referred to as $U(1) \rtimes \mathbb{Z}_4^{\mathsf{CT}}$ in the condensed matter literature.

Let us now contrast this with the conjectured infrared behavior of this theory [88]. The conjecture is that the theory has a massless Majorana fermion (the Goldstino) accompanied by a topological theory $U(N/2)_{N/2,N}$.

The Goldstino contributes a $\nu_{\text{goldstino}} = 1$ anomaly independently of the fractionalization class.[30] Now we need to consider the $\nu_{\text{T}}$ anomaly of $U(N/2)_{N/2,N}$, which is known to the $\pm 2$. Whether the TQFT $U(N/2)_{N/2,N}$ contributes 2 or $-2$ to the time-reversal anomaly depends on the fractionalization class. These results match with the UV computation since

$$\nu_{\text{IR}} = \nu_{\text{goldstino}} + \nu_{\text{TQFT}} = 1 \pm 2. \tag{96}$$

Let us quickly re-derive the statement that the TQFT $U(N/2)_{N/2,N}$ contributes $\pm 2$ to the time-reversal anomaly. We will use the conformal embedding picture.[31] Consider the conformal embedding [96]

$$U(n)_{k,k+n} \times U(k)_{n,k+n} \subset U(nk+1)_1. \tag{97}$$

The $U(nk+1)_1$ current algebra can be realized by $nk+1$ Dirac fermions $(\psi_{ia}, \Lambda)$, where $i = 1, \ldots, n$ and $a = 1, \ldots, k$, while the subalgebras are generated by

$$
\begin{aligned}
SU(n)_k &: \sum_a \psi_{ia}^* \psi_{ja} - \text{trace}, \\
SU(k)_n &: \sum_i \psi_{ia}^* \psi_{ib} - \text{trace}, \\
U(1)_{N(k+n)} &: \sum_{i,a} \psi_{ia}^* \psi_{ia} + N \Lambda^* \Lambda, \\
U(1)_{k(k+n)} &: \sum_{i,a} \psi_{ia}^* \psi_{ia} - k \Lambda^* \Lambda.
\end{aligned}
\tag{98}
$$

This embedding of chiral algebras induces the level-rank duality of Chern-Simons theories $U(n)_{k,k+n} \leftrightarrow U(k)_{-n,-k-n}$. When $n = k$ this duality yields the equivalence of $U(n)_{n,2n}$ and its time-reversal conjugate $U(n)_{-n,-2n}$. The operator that interchanges the two $U(n)_{n,2n}$ factors in (98) is

$$\text{T} : \psi_{ij} \to \pm \psi_{ji}, \qquad \text{T} : \Lambda \to \pm \Lambda^*. \tag{99}$$

The fermions that contribute to the anomaly are $\Lambda$ and thus $\nu_{\text{TQFT}} = \pm 2 \mod 16$. Note that in this particular example the two allowed values of the anomaly in the TQFT, $\pm 2$, are related both by changing the orientation and by changing the fractionalization class.

We will now consider the CT symmetry. We remind that this symmetry does not reverse the one-form symmetry charge, which will be important below.

The action of CT on the adjoint fermion can be taken to be

$$\text{CT} : \lambda_{ij} \to \gamma^0 \lambda_{ij}, \tag{100}$$

As discussed in section 2.3, we can compose CT with a matrix $U \in SU(N)$ that is either symmetric or anti-symmetric. The latter gives rise to a fractionalized action of CT, with anomaly $\nu_{\text{CT}} = -N - 1$, while the former to a non-fractionalized action, with $\nu_{\text{CT}} = +N - 1$. The shift

---

[30]The sign of $\nu_{\text{goldstino}}$ is fixed by the action of T on the supercurrent $S = \text{tr}(F\lambda)$, namely $\text{T} : S \mapsto +\gamma^0 S$. Thus, $\nu_{\text{goldstino}} = +1$.

[31]Examples of using such method to compute the 't Hooft anomaly for time-reversal symmetry in $2+1d$ TQFTs are discussed in [95].

is[32]

$$\Delta\nu_{\text{CT}} = 2N = \begin{cases} 0 & \text{mod } 16, \quad N = 0 \mod 8, \\ \pm 4 & \text{mod } 16, \quad N = \pm 2 \mod 8, \\ 8 & \text{mod } 16, \quad N = 4 \mod 8. \end{cases} \tag{101}$$

Let us now match the CT anomaly in the infrared "dual description" which consists of a Goldstino and the $U(N/2)_{N/2,N}$ TQFT. The anomaly in the infrared is[33]

$$\nu_{\text{IR}} = \nu_{\text{goldstino}} + \nu_{\text{TQFT}} = -1 + \nu_{\text{TQFT}}. \tag{102}$$

Anomaly matching, that is $\nu_{\text{CT}} = \nu_{\text{IR}}$, requires that $\nu_{\text{TQFT}} = \pm N \mod 16$. We now proceed to prove that indeed this is the case using the above conformal embedding. We identify the action CT for $\frac{N}{2} = n = k$ that exchanges the two $U(n)_{n,2n}$ factors in (97)

$$\text{CT} : \psi_{ij} \to \pm\psi_{ji}^*, \qquad \text{CT} : \Lambda \to \pm\Lambda. \tag{103}$$

The fermions that contribute to the anomaly are $\psi_{ii}$, and thus $\nu_{\text{TQFT}} = \pm 2n \mod 16$. This provides a new non-trivial consistency check of the proposed infrared dynamics for adjoint QCD [88] since the TQFT indeed contributes $\pm N \mod 16$ to the CT anomaly, as required by anomaly matching.

## 4.3 Other gauge groups and matter content

To check that the rules connecting the spin of the one-form symmetry generator line and the allowed anomaly jumps due to different fractionalization classes are indeed correct we have studied the effects of gauge transformations in many examples, summarized below, and found consistent results.

We denote $\theta = e^{2\pi i h}$, where $h$ is the spin of the generator of the one-form symmetry. We compute the time-reversal anomaly, following the same strategy as in section 2.2, for all rank-2 representations of the classical Lie groups. The naive anomaly $\nu_{\text{T}}$ is just the (real) dimension of the representation. On top of this, we can conjugate by a suitable gauge transformation $U \in G_{\text{gauge}}$. The shift $\Delta\nu$ equals $\nu_{\text{T}} - \nu_{\text{T}_U} \mod 16$. The only difference with section 2.2 is that now the anomaly takes values mod 16 instead of mod 8, and that T exists only when $T(R)$ is even, since the bare Chern-Simons level must be chosen as $k_{\text{b}} = T(R)/2$.

In what follows we quote the final result. Additional details may be found in section 3.8 of [97].

**Unitary group $SU(N)$.** Recall that the spin of $\mathbb{Z}_M \subseteq \mathbb{Z}_N$ is $h = \frac{kN}{M^2}(M-1)$. Here $M = N$ in the adjoint case, with $k = N/2$, or $M = 2$ in the (anti-)symmetric case, with $k = \frac{1}{2}(N \pm 2)$.

| $\Delta\nu_{\text{T}}$ | | | | $\theta$ | | | |
|---|---|---|---|---|---|---|---|
| $N = 0 \mod 8$ | $+4$ | $0$ | $0$ | $N = 0 \mod 8$ | $-i$ | $+1$ | $+1$ |
| $N = 2 \mod 8$ | $-4$ | $8$ | $0$ | $N = 2 \mod 8$ | $+i$ | $-1$ | $+1$ |
| $N = 4 \mod 8$ | $+4$ | $8$ | $8$ | $N = 4 \mod 8$ | $-i$ | $-1$ | $-1$ |
| $N = 6 \mod 8$ | $-4$ | $0$ | $8$ | $N = 6 \mod 8$ | $+i$ | $+1$ | $-1$ |

$$\tag{104}$$

---

[32]We note that when $N = 0 \mod 4$, some fractionalization class produces a two-group symmetry (see appendix F for details), where the background $B_2$ for the $\mathbb{Z}_2$ subgroup one-form symmetry obeys $dB_2 = w_1^3$, and thus the $\nu = 8$ anomaly as described the bulk term $\pi \int_{4d} w_1^4$ can be cancelled by a local counterterm $\pi \int_{3d} B_2 \cup w_1$ and becomes trivial.

[33]The sign of $\nu_{\text{goldstino}}$ is fixed by the action of CT on the supercurrent $S = \text{tr}(F\lambda)$, namely $\text{CT} : S \mapsto -\gamma^0 S$. Thus, $\nu_{\text{goldstino}} = -1$.

**Symplectic group $Sp(N)$.** Recall that the spin of $\mathbb{Z}_2$ is $h = \frac{1}{4}kN$. Here $k = \frac{1}{2}(N \pm 1)$.

| $\Delta \nu_{\mathsf{T}}$ | □□ | □ | | $\theta$ | □□ | □ |
|---|---|---|---|---|---|---|
| $N = 1 \mod 8$ | $-4$ | $0$ | | $N = 1 \mod 8$ | $+i$ | $+1$ |
| $N = 3 \mod 8$ | $8$ | $+4$ | | $N = 3 \mod 8$ | $-1$ | $-i$ |
| $N = 5 \mod 8$ | $+4$ | $8$ | | $N = 5 \mod 8$ | $-i$ | $-1$ |
| $N = 7 \mod 8$ | $0$ | $-4$ | | $N = 7 \mod 8$ | $+1$ | $+i$ |

$$(105)$$

**Orthogonal group $Spin(N)$.** Recall that for $N = 0 \mod 4$ we have a $\mathbb{Z}_4$ symmetry with spin $h = \frac{1}{16}kN$. For $N = 2 \mod 4$ we have a $\mathbb{Z}_2 \times \mathbb{Z}_2$ symmetry and the spins are $h = 0, \frac{1}{2}k, \frac{1}{16}kN, \frac{1}{16}kN$. Here, $k = \frac{1}{2}(N \pm 2)$.

| $\Delta \nu_{\mathsf{T}}$ | □□ | □ | | $\theta$ | □□ | □ |
|---|---|---|---|---|---|---|
| $N = 0 \mod 16$ | $8$ | $8$ | | $N = 0 \mod 16$ | $\pm 1$ | $\pm 1$ |
| $N = 2 \mod 16$ | $-4$ | $0$ | | $N = 2 \mod 16$ | $+1, +i$ | $+1$ |
| $N = 4 \mod 16$ | $+4, 8$ | $-4, 8$ | | $N = 4 \mod 16$ | $-1, -i$ | $-1, +i$ |
| $N = 6 \mod 16$ | $8$ | $+4$ | | $N = 6 \mod 16$ | $\pm 1$ | $+1, -i$ |
| $N = 8 \mod 16$ | $8$ | $8$ | | $N = 8 \mod 16$ | $-1$ | $-1$ |
| $N = 10 \mod 16$ | $+4$ | $8$ | | $N = 10 \mod 16$ | $+1, -i$ | $\pm 1$ |
| $N = 12 \mod 16$ | $-4, 8$ | $+4, 8$ | | $N = 12 \mod 16$ | $-1, +i$ | $-1, -i$ |
| $N = 14 \mod 16$ | $0$ | $-4$ | | $N = 14 \mod 16$ | $+1$ | $+1, +i$ |

$$(106)$$

In all these examples, it is clear that the shift $\Delta \nu$ is correlated with the spin of the generator of the one-form symmetry precisely as expected from the general considerations from before (cf. (85)). Namely, when the spin of the generator of the one-form symmetry is zero, the shift in the zero-form anomaly vanishes; when the spin is $1/2$, we find a shift of $\Delta \nu = 8$; and when the spin is $\mp 1/4$, the shift becomes $\Delta \nu = \pm 4$.

## 4.4 Magnetic symmetry: gauging the one-form symmetry

We close our study of $2 + 1d$ theories with a brief discussion of an apparent paradox, and its resolution. Consider a time-reversal invariant theory with one-form symmetry $\mathbb{Z}_2^{(1)}$. As above, this one-form symmetry may make the anomaly $\nu$ ambiguous with its corresponding $\Delta \nu$. A natural question one may ask is: what happens when we gauge $\mathbb{Z}_2^{(1)}$? Clearly, the resulting theory no longer has this one-form symmetry, and therefore $\nu$ should become unambiguous, with $\Delta \nu = 0$. How can we reconcile this with the ungauged theory having $\Delta \nu \neq 0$?

To be more precise, the symmetry $\mathbb{Z}_2^{(1)}$ can only be gauged when it does not have a pure one-form 't Hooft anomaly. This anomaly is measured by the spin of the generating line, namely $2h \mod 1$. Therefore, the one-form symmetry can only be gauged when the generating line is either a boson or a fermion. As discussed above, these two options lead to $\Delta \nu = 0$ and $\Delta \nu = 8$, respectively. The subtle case is the latter, since the ungauged theory has $\Delta \nu \neq 0$, while the gauged theory should have $\Delta \nu = 0$.

The resolution is the following. When we gauge $\mathbb{Z}_2^{(1)}$ we get a dual (magnetic) zero-form symmetry $\mathbb{Z}_2^{(0)}$, generated by a topological operator $\mathsf{M}$. The mixed anomaly between time-reversal and the one-form symmetry implies that, upon gauging the latter, the former gets extended [90, 98], and the algebra becomes

$$\mathsf{T}^2 = (-1)^F \mathsf{M}^{2h}. \tag{107}$$

This follows from the term $\pi \int B \cup w_1^2$, since the dual symmetry with background $B'$ couples as $\pi \int B \cup B'$, and the mixed anomaly can be cancelled by demanding $dB' = w_1^2$. This describes an extended T algebra, as specified by the element $w_1^2 \in H^2(\mathbb{Z}_4^{\mathsf{T}}, \mathbb{Z}_2) = \mathbb{Z}_2$.

This way, we see that when the generating line is a boson, the algebra is still $\mathsf{T}^2 = (-1)^F$, with unambiguous anomaly $\nu$. On the other hand, when the generating line is a fermion, the algebra becomes $\mathsf{T}^2 = (-1)^F \mathsf{M}$, and the anomaly is no longer given by a class $\nu \in \Omega^4_{\mathrm{Pin}^+}(\mathrm{pt}) = \mathbb{Z}_{16}$. Thus, it no longer makes sense to talk about $\Delta \nu$, since the anomaly of the extended algebra is given by an entirely different bordism group.

It would be interesting to repeat the analysis of this paper in theories with extended algebra $\mathsf{T}^2 = (-1)^F \mathsf{M}$, and the associated constraints on QCD phase diagrams.

# 5 QCD in 3+1 dimensions

## 5.1 Discrete axial zero-form symmetry and its anomaly

In this section we discuss $3 + 1d$ systems with the unitary $\mathbb{Z}_{2n}$ symmetry

$$g^n = (-1)^F, \tag{108}$$

with $(-1)^F$ being fermion number. The tangential structure describing this symmetry is

$$H = \frac{Spin \times \mathbb{Z}_{2n}}{\mathbb{Z}_2^F}. \tag{109}$$

The 't Hooft anomalies for this symmetry can be realized by free fermions, and they are classified by the following cobordism group [99, 100]

$$\Omega_5^H = \mathbb{Z}_a \times \mathbb{Z}_b, \tag{110}$$

where

$$a = \begin{cases} 24n & n = 0 \mod 6, \\ 8n & n \neq 0 \mod 3, n = 0 \mod 2, \\ 3n & n = 0 \mod 3, n \neq 0 \mod 2, \\ n & \text{otherwise}, \end{cases}$$

$$b = \begin{cases} n/6 & n = 0 \mod 6, \\ n/2 & n \neq 0 \mod 3, n = 0 \mod 2, \\ n/3 & n = 0 \mod 3, n \neq 0 \mod 2, \\ n & \text{otherwise}. \end{cases} \tag{111}$$

Thus, this symmetry is endowed with a large group anomalies.[34]

The symmetry (108) is ubiquitous in $3 + 1d$ gauge theories. Indeed, a gauge theory with gauge group $G_{\text{gauge}}$ and quarks in a (possibly reducible) representation $R$ of $G_{\text{gauge}}$ has the discrete symmetry (108) for some $n$. When $G_{\text{gauge}}$ is simply-connected, the symmetry is $\mathbb{Z}_{2T(R)}$, with $T(R)$ the Dynkin index of $R$. This symmetry is the unbroken subgroup of the classical chiral $U(1)$ symmetry that survives integrating over the $G_{\text{gauge}}$ gauge fields.

---

[34]An even larger group of anomalies can be studied by also considering charge conjugation, which satisfies

$$(g\mathsf{C})^2 = 1. \tag{112}$$

The extended symmetry is the dihedral group $D_{2n} \cong \mathbb{Z}_2 \ltimes \mathbb{Z}_{2n}$. Anomalies for $(Spin \times D_{2n})/\mathbb{Z}_2^F$ are less understood and therefore we shall stick to the $\mathbb{Z}_{2n}$ subgroup in what follows.

The anomaly can be computed by first embedding the $\mathbb{Z}_{2n}$ into $U(1)$, and then computing the anomaly perturbatively [99].[35] Let us denote the background $U(1)$ gauge field by $A'$. More precisely, due to the quotient in (109), it is a spin$^c$ connection. This means that on non-spin manifolds, the flux of $dA'$ is a multiple of $\pi$:

$$\frac{dA'}{2\pi} = \frac{1}{2}w_2(TM) \quad \mathrm{mod}\ \mathbb{Z}. \tag{115}$$

We restrict to the background field configuration such that $A'$ has holonomy in $\frac{2\pi}{2n}\mathbb{Z}$. Consider a collection of left-handed Weyl fermions with charges $q_i$ under the $\mathbb{Z}_{2n}$ symmetry, the anomaly can be described by the bulk effective action (see e.g. [102,103])

$$S_{\mathrm{anom}} = 2\pi \int_{5d} \left( \left(\sum_i q_i^3\right) \frac{1}{3!(2\pi)^3}A'dA'dA' - \left(\sum_i q_i\right) \frac{A'}{2\pi}\frac{1}{24}p_1(TM) \right), \tag{116}$$

which can be written as the boundary term of $2\pi \int_{6d} \mathrm{tr}\left(e^{dA'/2\pi}\hat{A}(R)\right)$ for Riemann curvature two-form $R$.

Since $A'$ has holonomy in $\mathbb{Z}_{2n}$, the anomaly takes discrete values. Let us consider the anomaly for a single Weyl fermion with unit charge,

$$I_1 = 2\pi \int_{5d} \left( \frac{1}{3!(2\pi)^3}A'dA'dA' - \frac{1}{24(2\pi)}A'p_1(TM) \right). \tag{117}$$

The bulk action $I_1$ is a multiple of $2\pi/(48n)$, and thus $kI_1$ for integer $k$ only depends on $k$ mod $48n$. If we consider 1 Weyl fermion of charge 3 and 27 Weyl fermions of charge $(-1)$, this gives a mixed global-gravity anomaly

$$I_2 = 2\pi \int_{5d} \frac{A'}{2\pi}p_1(TM). \tag{118}$$

The bulk action $I_2$ is a multiple of $2\pi/(2n)$, and thus $k'I_2$ for integer $k'$ only depends on $k'$ mod $2n$.

## 5.2 $SU(N)$ gauge theory with one adjoint fermion

We now proceed to study the dependence of these anomalies on a choice of fractionalization class. For definiteness, we consider $3+1d$ $SU(N)$ gauge theory with a quark $\psi$ in the adjoint representation, that is, $\mathcal{N} = 1$ $SU(N)$ super-Yang-Mills. This theory enjoys a $G = \mathbb{Z}_{2N}$ chiral zero-form symmetry as well as $\Gamma = \mathbb{Z}_N$ one-form symmetry. Therefore, the changes of fractionalization class take values in $H^2(\mathbb{Z}_{2N}, \mathbb{Z}_N) = \mathbb{Z}_N$.[36]

---

[35]The two generators for the anomaly can be chosen as follows. For $\mathbb{Z}_a$ it can be obtained from [99]

$$\nu_a = \frac{a}{48n}((2n^2 + n + 1)q_3 - (n+3)q_1) \quad \mathrm{mod}\ a, \tag{113}$$

where $q_k := \sum_i q_i^k$ and $a$ in (111). The anomaly for $\mathbb{Z}_b$ is known explicitly for $n$ a power of 2, in which case $b = n/2$. It reads [101]

$$\nu_b = \frac{b}{4n}((n+1)(2n+1)q_3 - (n+1)q_1) \quad \mathrm{mod}\ b. \tag{114}$$

We would like to thank Joe Davighi for discussions on this. In the following discussion, we will not need these generators.

[36]If we include the charge conjugation zero-form symmetry, then different fractionalization classes can produce a two-group for $N = 0$ mod 4 as discussed in appendix F.

The canonical $\mathbb{Z}_{2N}$ chiral symmetry acts on the $N^2 - 1$ adjoint fermions with charge $q_i = 1$, so that in $\mathcal{N} = 1$ $SU(N)$ super-Yang-Mills, the anomaly is described by the bulk effective action

$$S_{\text{anom}} = (N^2 - 1)I_1 = 2\pi(N^2 - 1)\int_{5d}\left(\frac{1}{3!(2\pi)^3}A'dA'dA' - \frac{1}{24(2\pi)}A'p_1(TM)\right). \quad (119)$$

We explore the dependence of the anomalies on the choice of fractionalization class by following the by-now familiar route of composing $g$ with gauge transformations $U$ such that $(gU)^N = (-1)^F$ on the fermions. For adjoint fields, the most general diagonal $SU(N)$ transformation with $\text{adj}(U^N) = 1$ is

$$U \propto \text{diag}\left(\mathbf{1}_{p_0}, \zeta\mathbf{1}_{p_1}, \zeta^2\mathbf{1}_{p_2}, \cdots, \zeta^{N-1}\mathbf{1}_{p_{N-1}}\right), \quad (120)$$

where $\zeta := e^{2\pi i/N}$, and $p_I$ are non-negative integers obeying $\sum_I p_I = N$. The global phase is fixed by unitarity, but it is irrelevant since it drops out when acting on fermions in the adjoint representation.

Collecting the components of $\psi$ into $p_I \times p_J$ blocks $\psi_{IJ}$, $U$ acts on each block as

$$\psi_{IJ} \mapsto e^{2\pi i(I-J)/N}\psi_{IJ}. \quad (121)$$

Therefore, the $\mathbb{Z}_{2N}$ charge of such a block is $q = 1 + 2(I - J)$, and

$$\sum q_i^k = -1 + \sum_{I,J=0}^{N-1} p_I p_J \left(1 + 2(I - J)\right)^k. \quad (122)$$

This allows us to compute the shifts of the charges of the fermions that enter in the computation of the anomalies

$$\Delta\left(\sum q_i\right) = -2\sum_{I,J=0}^{N-1} p_I p_J (I - J) = 0,$$

$$\Delta\left(\sum q_i^3\right) = -2\sum_{I,J=0}^{N-1} p_I p_J (I - J)(3 + 6(I - J) + 4(I - J)^2) \quad (123)$$

$$= -24N\sum_{I=0}^{N-1} p_I I^2 + 24\left(\sum_{I=0}^{N-1} p_I I\right)^2.$$

Thus we find that the anomaly changes by the bulk effective action

$$\Delta S_{\text{anom}} = 2\pi\Delta\left(\sum q_i^3\right)\int_{5d}\frac{1}{3!(2\pi)^3}A'dA'dA'$$

$$= 2\pi\left(\Delta\left(\sum q_i^3\right) \bmod 48N\right)\int_{5d}\frac{1}{3!(2\pi)^3}A'dA'dA', \quad (124)$$

where we used the property that the effective action takes value in $2\pi/(48N)$. Let us denote

$$\ell \equiv \sum_I I p_I. \quad (125)$$

Since

$$\Delta\left(\sum q_i^3\right) \bmod 48N = -24N\left(\sum_I p_I I\right) + 24\ell^2 \bmod 48N$$

$$= 24(-N + 1)\ell^2 \bmod 48N, \quad (126)$$

we find the anomaly changes by

$$\Delta S_{\text{anom}} = 2\pi \cdot 24(-N+1)\ell^2 \int_{5d} \frac{1}{3!(2\pi)^3} A' dA' dA'$$

$$= \frac{(-N+1)\ell^2}{\pi^2} \int_{5d} A' dA' dA' \tag{127}$$

$$= 24(-N+1)\ell^2 I_1 + (-N+1)\ell^2 I_2 \,.$$

## 5.3  Different fractionalization classes, and one-form symmetry anomaly

$SU(N)$ gauge theory with one adjoint fermion has $\mathbb{Z}_N$ one-form symmetry. The one-form symmetry has a mixed anomaly with the $\mathbb{Z}_{2N}$ axial zero-form symmetry. Under an axial rotation labelled by $k \in \mathbb{Z}_{2N}$

$$\psi_i \to \psi_i e^{\frac{2\pi i k}{2N}} \,. \tag{128}$$

The Adler-Bell-Jackiw anomaly [104, 105] implies that the $\theta$ angle is shifted by

$$\theta \to \theta + 2\pi k \,. \tag{129}$$

In the presence of background gauge field $B$ for the $\mathbb{Z}_N$ one-form symmetry that changes the $\theta$ periodicity, the transformation does not leave the theory invariant, but produces [26, 81]

$$2\pi \frac{k(N-1)}{2N} \int_{4d} \mathcal{P}(B) \,. \tag{130}$$

Thus there is a mixed anomaly between the chiral symmetry and the one-form symmetry. The anomaly is described by the $4 + 1d$ effective action (the overall sign depends on the orientation)[37]

$$-2\pi \frac{N-1}{2N} \int_{5d} \mathcal{P}(B) \cup A \quad \text{mod } 2\pi \mathbb{Z} \,, \tag{132}$$

where $A$ is the background gauge field for the $\mathbb{Z}_{2N}$ chiral symmetry, that satisfies

$$dA = Nw_2(TM) \quad \text{mod } 2N \,. \tag{133}$$

Different fractionalization classes are obtained by turning on a background[38]

$$B = \ell \frac{dA}{N} \text{ mod } N \,, \quad \ell = 0, 1, \cdots, N-1 \,. \tag{134}$$

---

[37]For odd $N$, the action is a multiple of $2\pi/N$, and since $dA = 0$ mod $N$, the action is well-defined. For even $N$, the action is a multiple of $2\pi/(2N)$, and it can be shown to be well-defined by extending it to the bulk

$$-2\pi \frac{N-1}{2N} \int_{6d} \mathcal{P}(B) \cup dA = -\pi \int B \cup B \cup w_2(TM) = -\pi \int_{6d} Sq^1(B) Sq^1(B)$$

$$= \pi \int_{6d} Sq^1(B Sq^1(B)) = \pi \int_{6d} w_1(TM) B Sq^1(B) \text{ mod } 2\pi \,, \tag{131}$$

which is trivial for orientable manifolds, where $w_1(TM) = 0$, and we used $Sq^2(B \cup B) = Sq^1(B) Sq^1(B)$ and $Sq^k x_{d-k} = v_k(TM) \cup x_{d-k}$ for $\mathbb{Z}_2$ $(d-k)$ cocycle on $d$-dimensional manifold [86]. Thus the action is well-defined in $4 + 1d$.

[38]Following the same analysis as in section 2.2, we learn that the fractionalization class induced by $U$ is given by the phase of $U^N$, which equals $U^N = \zeta^{-\ell} \mathbf{1}_N$ and only depends on the combination $\ell = \sum_I p_I I$. This implies that the change in the fractionalization class (and thus the change in the anomaly) only depends on the integers $p_I$ via the combination $\ell = \sum_I I p_I$.

We note that for even $N$, $\ell$ can be $N/2$, then the corresponding fractionalization class is $B = \frac{N}{2} w_2(TM)$ mod $N$. For fractionalization class $\ell$, the mixed one-form/zero-form anomaly gives additional anomaly for the zero-form symmetry

$$-2\pi \frac{(N-1)\ell^2}{2N} \int_{5d} \mathcal{P}\left(\frac{dA}{N}\right) \cup A. \tag{135}$$

Written in terms of $A' = \frac{2\pi}{2N}A$, this is

$$-2\pi \frac{(N-1)\ell^2}{2N} \int_{5d} \left(\frac{dA'}{\pi}\right)^2 \frac{2N}{2\pi} A' = \frac{(-N+1)\ell^2}{\pi^2} \int_{5d} A' dA' dA'. \tag{136}$$

Thus changing the fractionalization classes reproduces the jump of the anomaly as in (127).

## 5.4 Domain walls

In this section we make a few comments about the matching of the $\mathbb{Z}_{2n}$ anomaly by the low-energy effective theory. We consider $\mathcal{N} = 1$ super-Yang-Mills, that is, adjoint QCD with some gauge group $G_{\text{gauge}}$. In this case, the axial symmetry is $\mathbb{Z}_{2h}$, where $h = T(\text{adj})$ is the Coxeter number of $G_{\text{gauge}}$.

It is a well-known fact that $3 + 1d$ $\mathcal{N} = 1$ super-Yang-Mills with simply-connected gauge group is gapped and confining, and that the infrared consists of $h$ discrete vacua as a result of the spontaneous breaking of the $\mathbb{Z}_{2h}$ symmetry down to $\mathbb{Z}_2^F$. These vacua are separated by domain walls, which carry non-trivial $2 + 1d$ degrees of freedom. Specifically, the wall separating $p \in \mathbb{Z}_h$ units of vacua supports a $2 + 1d$ $\mathcal{N} = 1$ super-Yang-Mills theory with the same gauge group and a Chern-Simons level $k = h/2 - p$ [92, 106]. It is these degrees of freedom what matches the $\mathbb{Z}_{2h}$ anomaly as computed in the ultraviolet.

For simplicity, here we will restrict ourselves to the parity-symmetric wall $p = h/2$ that exists when $h$ is even. This wall is special since the CPT wall construction [66] implies that the unitary axial symmetry in the bulk is transmuted to an anti-unitary symmetry on the wall. In this case, it is very easy to show that the ultraviolet anomaly is matched by the $3d$ degrees of freedom.

The time-reversal symmetric wall corresponds to the $\mathbb{Z}_2$ subgroup of $\mathbb{Z}_h$, i.e., to a spontaneously broken $g^2 = (-1)^F$ symmetry. In the bulk, anomalies for such unitary symmetry are classified by $\mathbb{Z}_a \times \mathbb{Z}_b$, where $a, b$ are given by (111) at $n = 2$, namely $a = 16$ and $b = 1$. In other words, the anomalies take values in $\mathbb{Z}_{16}$. On the wall, this a time-reversal symmetry, whose anomalies are also classified by $\mathbb{Z}_{16}$. It is easy to show that these anomalies indeed match.

The $\mathbb{Z}_{16}$ anomaly in the bulk is given by (113) with $n = 2$, to wit,

$$\nu_a = \frac{11q_3 - 5q_1}{6} \mod 16. \tag{137}$$

If we let $n_\pm$ be the number of fermions that transform with sign $\pm 1$, then $q_k = n_+ - n_-$, and the anomaly simplifies to

$$\nu_a = n_+ - n_- \mod 16, \tag{138}$$

which precisely matches the time-reversal anomaly $\nu_T$ of the wall, namely the time-reversal anomaly of $3d$ adjoint QCD.

The non-parity-symmetric walls require a more careful study that we leave to future work. In this case, non-invertible topological defects are potentially important, as well as junctions of domain walls. See [70, 107–116] for related work in various dimensions.

## Acknowledgements

We thank M. Cheng, J. Davighi, C.-M. Jian, N. Seiberg, T. Senthil and M. Yu for discussions. We thank Maissam Barkeshli, Anton Kapustin, Nathan Seiberg, Shu-Heng Shao, Yuya Tanizaki, Juven Wang and Cenke Xu for comments on a draft. P.-S.

**Funding information**   H. is supported by the Simons Collaboration on Global Categorical Symmetries. D.D. and J.G. acknowledge support by Perimeter Institute for Theoretical Physics. Research at Perimeter Institute is supported in part by the Government of Canada through the Department of Innovation, Science and Economic Development Canada and by the Province of Ontario through the Ministry of Colleges and Universities. Any opinions, findings, and conclusions or recommendations expressed in this material are those of the authors and do not necessarily reflect the views of the funding agencies.

## A   Congruence vector of simple Lie algebras

In this appendix we quote the value of $\rho \cdot v \mod |Z(G_{\text{gauge}})|$ for the classical Lie groups, where $\rho = (1, 1, \ldots, 1)$ is the Weyl vector.

**Unitary group $SU(N)$.**   The congruence vector is $v = (1, 2, \ldots, N-1)$. Thus,

$$\rho \cdot v = \frac{1}{2}N(N-1) \mod N, \tag{A.1}$$

which is nonzero when $N$ is even.

**Symplectic group $Sp(N)$.**   The congruence vector is $v = (1, 2, \ldots, N)$. Thus,

$$\rho \cdot v = \frac{1}{2}N(N+1) \mod 2, \tag{A.2}$$

which is nonzero when $N = 1, 2 \mod 4$.

**Orthogonal group $SO(N)$.**   The congruence vectors depend on $N \mod 4$, which we consider in turn.
   • In $SO(2n+1)$, the congruence vector is $v = (0, 0, \ldots, 1)$. Thus,

$$\rho \cdot v = 1 \mod 2, \tag{A.3}$$

which is always nonzero.
   • In $SO(4n+2)$, the congruence vector is $v = (2, 4, 6, \ldots, -2n+1, -2n-1)$. Thus,

$$\rho \cdot v = -2n \mod 4, \tag{A.4}$$

which is nonzero when $n$ is odd.
   • In $SO(4n)$, the congruence vectors are $v_1 = (0, 0, \ldots, 1, 1)$ and $v_2 = (1, 2, 3, \ldots, 1-n, -n)$. Thus,

$$\begin{aligned} \rho \cdot v_1 &= 0 \mod 2, \\ \rho \cdot v_2 &= -n \mod 2, \end{aligned} \tag{A.5}$$

and the latter is nonzero when $n$ is odd.
   These results are summarized in table (19).

# B One-form symmetry anomaly in QCD$_{0+1}$

Consider QCD in $0+1d$ with simple gauge group $G_{\text{gauge}}$ and a fermion in some representation $R$. The one-form symmetry $\Gamma$ is the subgroup of the center of gauge group that acts trivially on the representation $R$. We would like to compute the 't Hooft anomaly of the one-form symmetry.

Since the background for the one-form symmetry changes the $G_{\text{gauge}}$ gauge field into $G_{\text{gauge}}/\Gamma$ gauge field, to address the question we can compute the partition function of the free fermion theory in the presence of a background gauge field for $G_{\text{gauge}}/\Gamma$. While the partition function with background $G_{\text{gauge}}$ is well-defined (and thus the QCD$_{0+1}$ with the corresponding gauge group is free of gauge anomalies), the partition function with background $G_{\text{gauge}}/\Gamma$ can have ambiguities, which is the anomaly of the one-form symmetry $\Gamma$ in QCD$_{0+1}$ with gauge group $G_{\text{gauge}}$. Such anomaly arises if and only if the symmetry $G_{\text{gauge}}/\Gamma$ is realized projectively on the Hilbert space of the free fermions.

Consider the embedding of $\mathfrak{g}_{\text{gauge}}$ inside $\mathfrak{so}(\dim(R))$ via the representation $R$. This embedding is defined by the branching rule

$$
\begin{array}{rcl}
\mathfrak{so}(\dim(R)) & \supseteq & \mathfrak{g}_{\text{gauge}} \\
\square & \mapsto & R\,.
\end{array}
\tag{B.1}
$$

Let us first turn on a holonomy $U$ of the background gauge field that takes value in $Spin(\dim(R))$. Since the Hilbert space realizes a spinor representation $s$ of $Spin(\dim(R))$, the partition function can be expressed in terms of the character of such spinor representation:

$$
\begin{aligned}
Z_\pm(U) &= \chi_s(U) \pm \chi_{\bar{s}}(U), & \dim(R) \text{ even}, \\
Z_+(U) &= \sqrt{2}\chi_s(U), & \dim(R) \text{ odd}, \\
Z_-(U) &= 0, & \dim(R) \text{ odd}.
\end{aligned}
\tag{B.2}
$$

where $\pm$ denotes the spin structure (Ramond or Neveu-Schwarz), and $s$ the spinor representation of $Spin(\dim(R))$ (and $\bar{s}$ the conjugate representation). When we restrict the holonomy $U$ to take value in $G_{\text{gauge}}$, the partition function is

$$
\begin{aligned}
Z_\pm(U) &= \chi_{s(R)}(U) \pm \chi_{\bar{s}(R)}(U), & \dim(R) \text{ even}, \\
Z_+(U) &= \sqrt{2}\chi_{s(R)}(U), & \dim(R) \text{ odd}, \\
Z_-(U) &= 0, & \dim(R) \text{ odd}.
\end{aligned}
\tag{B.3}
$$

where $s(R) \in \text{Rep}(G_{\text{gauge}})$ is the image of $s \in \text{Rep}(Spin(\dim(R)))$ under the embedding (B.1),

$$
\begin{array}{rcl}
\mathfrak{so}(\dim(R)) & \supseteq & \mathfrak{g}_{\text{gauge}} \\
s & \mapsto & s(R)\,.
\end{array}
\tag{B.4}
$$

(In general, $s(R)$ is a reducible representation even if $R$ is irreducible.) The partition function without insertion transforms under the center $\Gamma$ according to the transformation of $s(R)$. Thus in the $G_{\text{gauge}}$ gauge theory, the one-form symmetry has anomaly if and only if $s(R)$ transforms non-trivially under $\Gamma$ in the center of $G_{\text{gauge}}$. The anomaly can be described by the bulk term

$$
\int_{2d} \langle q(s(R)), B \rangle\,,
\tag{B.5}
$$

where $q(s(R)) \in \text{Hom}(\Gamma, U(1))$ is the one-form charge of $s(R)$ under $\Gamma$ in the center of $G_{\text{gauge}}$, and $B$ is the background two-form gauge field for the $\Gamma$ one-form symmetry.

As an application of this result, let us show that fermions in complex representations do not contribute to the anomaly of the one-form symmetry. When $R$ is complex, the branching rules are

$$s(R) \mapsto \bigoplus_{j=0}^{\lfloor \dim(R)/2 \rfloor} \wedge^{2j} R \,,$$

$$\bar{s}(R) \mapsto \bigoplus_{j=0}^{\lceil \dim(R)/2 \rceil - 1} \wedge^{2j+1} R \,. \tag{B.6}$$

Of course, $\wedge^n R$ is not charged under $\ker(R)$ for any $n$, and therefore complex representations never lead to an anomaly for one-form symmetry.

Anomalies are only present in the case of real representations. As a simple illustration, consider adjoint QCD. Here the embedding becomes $\mathfrak{so}(\dim G_{\text{gauge}}) \supset \mathfrak{g}_{\text{gauge}}$, and the spinor representation branches as

$$s \mapsto 2^{\lceil r/2 \rceil - 1} \rho \,, \tag{B.7}$$

where $\rho$ is the Weyl vector of $\mathfrak{g}_{\text{gauge}}$ and $r = \text{rank}(\mathfrak{g}_{\text{gauge}})$. Therefore, the one-form symmetry of adjoint QCD is anomalous if and only if $\rho$ is charged under it, in agreement with section 2.1.

An equivalent way to phrase this discussion is in terms of how exactly $G_{\text{gauge}}$ embeds into $Spin(\dim(R))$. We sketch the argument for adjoint $SU(N)$ below.

## B.1  $SU(N)$ gauge theory with an adjoint fermion

Consider $SU(N)$ gauge theory with an adjoint fermion in $0+1d$. We would like to understand whether the $\mathbb{Z}_N$ one-form symmetry is anomalous.

This is equivalent to asking whether the $PSU(N)$ zero-form symmetry that acts on $N^2 - 1$ free Majorana fermions is anomalous. In other words, whether the Hilbert space of the free fermions transforms as projective representation of $PSU(N)$ symmetry. When $N$ is odd, there is an even number of Majorana fermions and a properly $\mathbb{Z}_2$-graded Hilbert space; when $N$ is even, we add one extra singlet Majorana fermion to form a well-defined Hilbert space.

The discussion divides into two cases: $N$ even and $N$ odd.

### B.1.1  Even $N$

When $N$ is even, we add one additional Majorana fermion that is singlet under $PSU(N)$ to obtain a well-defined Hilbert space. We quantize the fermions, which form the Clifford algebra of rank $N^2$. The Hilbert space forms the spinor representation of $Spin(N^2)$ symmetry. The states generated by the $N^2 - 1$ Majorana fermions form spinor of $Spin(N^2 - 1)$. We have the embedding

$$Spin(N^2 - 1) \subset SU(N)/\mathbb{Z}_{N/2} \,, \tag{B.8}$$

where both sides have $\mathbb{Z}_2$ center. Thus the spinor representation decomposes into $SU(N)$ representation with non-trivial $N$-ality, and thus the $PSU(N)$ symmetry acts projectively on the Hilbert space. This means the $PSU(N)$ symmetry is anomalous, with the anomaly

$$\frac{2\pi(N/2)}{N} \int w_2^{PSU(N)} = \pi \int w_2^{PSU(N)} \,. \tag{B.9}$$

We note that this implies that the Hilbert space of $SU(N)$ gauge theory with adjoint fermion is empty (since the vacuum carries non-trivial $SU(N)$ representation) unless considering the twisted Hilbert space with insertion of Wilson line of $N$-ality $N/2$ in the time direction.

In the $SU(N)$ gauge theory with adjoint fermion, this implies the one-form symmetry is anomalous. In terms of the one-form symmetry gauge field $B_2$, the anomaly is

$$\pi \int B_2. \tag{B.10}$$

### B.1.2 Odd $N$

When $N$ is odd, $N^2 - 1$ is even, and there is a well-defined Hilbert space. Let us quantize the fermions, which give Clifford algebra of rank $N^2 - 1$. The Hilbert space forms the spinor representations of $Spin(N^2 - 1)$. Let us study how the Hilbert space transforms under $SU(N)$ symmetry. Since $SU(N)$ for odd $N$ does not have a $\mathbb{Z}_2$ subgroup center, the spinor representation of $Spin(N^2 - 1)$ decompose into $SU(N)$ representation with 0 $N$-ality, and thus we have the embedding

$$Spin(N^2 - 1) \supset PSU(N). \tag{B.11}$$

Thus the $PSU(N)$ symmetry acts linearly on the Hilbert space, and it does not have an anomaly. Thus we conclude that when $N$ is odd, the $\mathbb{Z}_N$ one-form symmetry in $SU(N)$ gauge theory with one adjoint fermion is non-anomalous.

## C $\quad U(1)$ gauge theory in 1+1d

In this appendix we use one-form symmetry to discuss the anomaly of $\mathbb{Z}_2$ charge conjugation symmetry in $U(1)$ gauge theory in $1 + 1d$ for different fractionalization classes.

Consider the $U(1)$ gauge theory with $\theta = \pi$

$$\frac{1}{2e^2} da \star da + \frac{1}{2} da. \tag{C.1}$$

The theory has $\Gamma = U(1)$ one-form symmetry that shifts $a$, and $\mathbb{Z}_2$ charge conjugation symmetry that acts as $a \to -a$.

Let us compute the anomaly for the symmetries. We turn on background $B_2$ for the $U(1)$ one-form symmetry, and $B_1$ for the $\mathbb{Z}_2$ charge conjugation symmetry. This replaces $da$ by

$$D_{B_1} a - B_2 = (d - 2B_1)a - B_2. \tag{C.2}$$

Thus the theta term contributes the bulk dependence

$$\int \frac{1}{2}(2B_1 da - dB_2) = \int B_1 B_2 - \frac{1}{2}\int dB_2 = -\frac{1}{2}\int D_{B_1} B_2. \tag{C.3}$$

This is a mixed anomaly between the one-form symmetry and charge conjugation zero-form symmetry.

There are two fractionalization classes for the $\mathbb{Z}_2$ charge conjugation 0-form symmetry, classified by $H^2(B\mathbb{Z}_2, \mathbb{Z}_2) = \mathbb{Z}_2$:

- $B_2 = 0$, then the zero-form symmetry is not anomalous.

- $B_2 = \pi B_1^2$, then the zero-form symmetry has an anomaly

$$\pi \int B_1^3. \tag{C.4}$$

This agrees with the discussion in section 3.

# D  Fermionic $\mathbb{Z}_2$ gauge theory with $\mathbb{Z}_2$ symmetry in 1+1d

Let us consider $\mathbb{Z}_2$ gauge theory in $1+1d$, viewed as a fermionic theory, and enriched with unitary $\mathbb{Z}_2 \times \mathbb{Z}_2^F$ symmetry, where $\mathbb{Z}_2^F$ is the fermion parity symmetry.

The action without any background gauge fields is

$$\pi \int a_0 \cup \delta b_1 \,. \tag{D.1}$$

If we integrate out $b_1$, this sets $\delta a_0 = 0 \bmod 2$. Such condition will be modified in the presence of background gauge fields.

The theory has $\mathbb{Z}_2$ one-form symmetry that acts on the Wilson line, and different symmetry fractionalizations in the fermionic world admit a classification using the spin bordism group

$$\mathrm{Hom}(\Omega_2^{spin}(B\mathbb{Z}_2), \mathbb{Z}_2) \,. \tag{D.2}$$

The group is $\mathbb{Z}_2 \times \mathbb{Z}_2$, generated by the Arf invariant, and the quadratic function $\frac{1}{2}q_\rho(A)$, where we denote the spin structure by $\rho$, and the $\mathbb{Z}_2$ background gauge field by $A$. If we denote the symmetry fractionalization by $(m,n) \in \mathbb{Z}_2 \times \mathbb{Z}_2$, then the path integral is modified to be

$$Z = \int Da_0 \, e^{\frac{2\pi i m}{8} \mathrm{Arf}(\mathrm{PD}_{2d}(a_0))} e^{n\frac{\pi i}{2} \int_{\mathrm{PD}_{2d}(\phi_0)} q_\rho(A)} \,, \tag{D.3}$$

where $\mathrm{PD}_{2d}(\phi_0)$ is the two-cycle that is Poincaré dual to $\phi_0$. It is closed in $\mathbb{Z}_2$ homology, since $\delta\phi_0 = 0 \bmod 2$, and thus the above partition function is well-defined. We note that $\mathrm{PD}_{2d}(\phi_0)$ is orientable,[39] and thus $q_\rho$ is even, and Arf is a multiple of 4 in the above normalization, and the phases in the path integral are signs. We can also include $(-1)^{\ell \int_{\mathrm{PD}(\phi_0)} A \cup A}$ for $\ell \in \mathbb{Z}_2$, which is trivial for $\delta\phi_0 = 0$.

The theory also has $\mathbb{Z}_2$ zero-form symmetry that acts on the $\mathbb{Z}_2$ vortex point operator. Let us consider the $\mathbb{Z}_2$ unitary symmetry that acts on such operators. Then, in the presence of background $A$, $\phi_0$ is no longer closed in $\mathbb{Z}_2$,

$$\delta\phi_0 = A \,, \tag{D.4}$$

and $\mathrm{PD}_{2d}(\phi_0)$ has a boundary that is non-trivial in $\mathbb{Z}_2$ homology

$$\partial \mathrm{PD}_{2d}(\phi_0) = \mathrm{PD}_{2d}(d\phi_0) = \mathrm{PD}_{2d}(A) \,. \tag{D.5}$$

In such a case, the partition function is no longer well-defined. This is a manifestation of an anomaly for the $\mathbb{Z}_2$ symmetry. One way to cure the problem is by introducing a $3d$ bulk, and attaching the boundary surface to another surface in the bulk that anchors on the boundary along $\mathrm{PD}_{2d}(A)$, to form a closed surface, so the phases in the partition function evaluated on the 2-cycle $\Sigma_{\text{2-cycle}}$ can be defined, where

$$\Sigma_{\text{2-cycle}} = \mathrm{PD}_{2d}(\phi_0) \cup \overline{\Sigma_{\text{bulk}}} \,, \quad \partial\Sigma_{\text{bulk}} = \mathrm{PD}_{2d}(A) \,. \tag{D.6}$$

For instance, if $2d$ spacetime has trivial topology and we take the bulk direction to be the $z$ coordinate, then such a bulk surface can be a cigar with base $\mathrm{PD}_{2d}(A)$ and extends along the $z$ direction.

If we consider two bulks, the choice of bulk surface $\Sigma_{\text{bulk}}$ depends on $\mathrm{PD}_{3d}(A)$, which is a surface in the bulk (note it may be an unorientable surface embedded in an orientable bulk,

---

[39]For a $2d$ surface $D$ embedded in 2d orientable spacetime $M$, $TM| = TD + ND$, but $ND = 0$ since it has zero dimension, and thus $TM| = TD$, and for orientable $TM$, $TD$ is also orientable.

if $A$ has a non-trivial bundle in the bulk). When the bulk surface is unorientable, we need to refine the Arf invariant into ABK invariant. Then the phases depend on the bulk by

$$e^{\frac{2\pi i m}{8}\text{ABK}(\text{PD}_{3d}(A))}e^{n\frac{\pi i}{2}\int_{\text{PD}_{3d}(A)}q_\rho(A)}(-1)^{\ell\int_{\text{PD}_{3d}(A)}A\cup A}, \tag{D.7}$$

where the first term is the partition function for the root phase of the $\mathbb{Z}_8$ invertible phases in $2+1d$ with $\mathbb{Z}_2\times\mathbb{Z}_2^F$ symmetry and zero thermal Hall conductance. The second term can be written as an odd-level Chern-Simons term for the gauge field $A$. The third term is the partition function for the bosonic SPT phase with $\mathbb{Z}_2$ symmetry. The above decomposition represents the Arf layer, $\psi$ layer and bosonic layer in [58]. In particular, $m, n$ now have extended range, with the identification

$$(m+2,n,\ell)\sim(m,n+1,\ell),\quad(m,n+2,\ell)\sim(m,n,\ell+1). \tag{D.8}$$

# E  $\mathbb{Z}_2$ gauge theory with two-group symmetry in 2+1d

## E.1  Intrinsic symmetries and their anomalies

Untwisted $\mathbb{Z}_2$ gauge theory in $2+1d$ has $\mathbb{Z}_2\times\mathbb{Z}_2$ one-form symmetry generated by the electric and magnetic line operators. The theory also has unitary $\mathbb{Z}_2^C$ zero-form symmetry that exchanges the electric and magnetic lines. Let us denote the background for zero-form symmetry by $B_1$, the background for the diagonal $\mathbb{Z}_2$ one-form symmetry generated by the dyon by $B_2$, and the background for the $\mathbb{Z}_2$ one-form symmetry generated by the Wilson line by $B_2'$. They obey the twisted cocycle condition

$$dB_2 = B_1\cup B_2',\quad dB_2' = 0,\quad dB_1 = 0. \tag{E.1}$$

To understand this relation, we note that in the presence of backgrounds $B_2', B_1$, there must also be a non-trivial background for $B_2$ due to the action of the zero-form symmetry on the one-form symmetry, and this is captured by the first relation.

More generally, we can consider $Spin(4k)_1$ Chern-Simons theory for integer $k$, which has $\mathbb{Z}_2\times\mathbb{Z}_2$ one-form symmetry generated by the lines in the spinor and cospinor representation, and $\mathbb{Z}_2$ charge conjugation 0-from symmetry that exchanges the two lines. The Toric Code $\mathbb{Z}_2$ gauge theory corresponds to $k=0$.

In addition, the theory has $\mathbb{Z}_2^T$ time-reversal that does not permute the line operators. Let us turn on background field by placing the theory on an unorientable manifold with first Stiefel-Whitney class $w_1$. Then the twisted cocycle condition is modified to be [117]

$$dB_2 = B_1\cup\left(B_2' + w_1\cup B_1\right). \tag{E.2}$$

We note that under a $\mathbb{Z}_2$ one-form transformation for $B_2'$, $B_2'\to B_2'+d\lambda_1$, the background $B_2$ is shifted by $B_2\to B_2+B_1\cup\lambda_1$, where we used $dB_1=0$. When later we focus on unitary $\mathbb{Z}_2^C$ zero-form symmetry or time-reversal symmetry that acts as charge conjugation, this implies that the non-trivial fractionalization classes only come from shifting $B_2'$ but not $B_2$.

The zero-form and one-form symmetries have an anomaly, described by the bulk term [117]

$$S_{\text{bulk SPT}} = \pi\int\mathcal{P}(B_2) + \pi\int B_2\cup(B_2'+w_1\cup B_1) + \pi\int w_1\cup B_1\cup B_2$$

$$+\pi\int B_1\cup\left[(B_1\cup(B_2'+w_1\cup B_1))\cup_2(B_2'+w_1\cup B_1) + B_1\cup(B_2'+w_1\cup B_1)\right]$$

$$+\frac{\pi}{2}\int B_1^2\cup(B_2'+w_1\cup B_1), \tag{E.3}$$

where in the last term we take a lift of $B_1$ to $\mathbb{Z}_4$ cochain that takes value in $0,1$.

## E.2 Unitary $\mathbb{Z}_2^{\mathsf{C}}$ symmetry

Let us consider unitary $\mathbb{Z}_2^{\mathsf{C}}$ symmetry, with $w_1 = 0$. The twisted cocycle condition for $B_2, B_2'$ becomes

$$dB_2 = B_1 \cup B_2'. \tag{E.4}$$

We note that the global transformation $B_2' \to B_2' + dB_1 = B_2'$ induces the equivalence relation $B_2 \sim B_2 + B_1^2$.

Let us consider the following symmetry fractionalizations:

- $B_2' = 0$: charge conjugation squares to $+1$ on the dyon. Then $dB_2 = 0$, and the zero-form symmetry does not participate in a non-trivial two-group with the one-form symmetry.

- $B_2' = B_1^2$: charge conjugation squares to $-1$ on the dyon. Then $dB_2 = B_1^3$, the zero-form symmetry participates in a two-group with the one-form symmetry, with Postnikov class $B_1^3$.

### E.2.1 Anomaly

The zero-form and one-form symmetry has an anomaly, described by the bulk term [117]

$$\pi \int \mathcal{P}(B_2) + \pi \int B_2 \cup B_2' + \pi \int B_1 \cup \big( (B_1 \cup B_2') \cup_2 B_2' + B_1 \cup B_2' \big) + \frac{\pi}{2} \int B_1^2 \cup B_2', \tag{E.5}$$

where in the last term we take a lift of $B_1$ to $\mathbb{Z}_4$ cochain that takes value in $0, 1$.

Thus for $B_2' = 0$, the zero-form symmetry by itself does not have an anomaly.

For $B_2' = B_1^2$, there is a 2-group anomaly:

$$\pi \int \mathcal{P}(B_2) + \pi \int B_2 \cup B_1^2 + \pi \int B_1 \cup \big( (B_1^3) \cup_2 B_1^2 \big) + \frac{3\pi}{2} \int B_1^4. \tag{E.6}$$

The last term $B_1^4$ can be canceled by a local $3d$ counterterm, and we are left with the first three terms.[40] Thus the $\mathbb{Z}_2$ gauge theory gives an example with anomalous 2-group symmetry.

## E.3 Anti-unitary $\mathbb{Z}_2$ symmetry that acts as charge conjugation

Let us consider instead $\mathbb{Z}_2$ time-reversal symmetry that acts as charge conjugation symmetry, i.e. the diagonal $\mathbb{Z}_2$ subgroup zero-form symmetry for $\mathbb{Z}_2^{\mathsf{C}} \times \mathbb{Z}_2^{\mathsf{T}}$. In other words, we restrict to $B_1 = w_1$. Then the twisted cocycle conditions for $B_2, B_2'$ becomes

$$dB_2 = w_1 \cup \big( B_2' + w_1^2 \big). \tag{E.7}$$

We note that the global transformation $B_2' \to B_2' + dw_1 = B_2'$ induces the equivalence relation $B_2 \sim B_2 + w_1^2$.

Let us consider the following symmetry fractionalization for the $\mathbb{Z}_2$ anti-unitary symmetry:

- $B_2' = 0$: time-reversal squares to $+1$ on the dyon. Then $dB_2 = w_1^3$, and the zero-form symmetry participates in a two-group with the one-form symmetry, with Postnikov class $w_1^3$.

- $B_2' = w_1^2$: time-reversal squares to $-1$ on the dyon. Then $dB_2 = 0$, and the zero-form symmetry does not participate in a non-trivial two-group with the one-form symmetry.

---

[40]To see the anomaly is well-defined, we can extend it to $5d$, and use $\pi \int B_1^5 = \pi Sq^2(B_1^2) \cup B_1 = \pi \int Sq^2(B_1^3) = \pi \int d\mathcal{P}(B_2) + \pi \int d\zeta(B_1, B_1^2)$, where $\zeta(B_1, B_1^2)$ is the Cartan coboundary given by the term that depends only on $B_1$, and we used $d\mathcal{P}(B_2) = dB_2 \cup_1 dB_2 = Sq^2(B_1^3)$ mod 2 with $dB_2 = B_1^3$.

The condition that only certain fractionalization class leads to a symmetry that does not participate in a two-group is also discussed in equation (234) of [40]. See also [118] for related examples.[41]

### E.3.1  Anomaly

The zero-form and one-form symmetries have an anomaly, described by the bulk term [117]

$$S_{\text{bulk SPT}} = \pi \int \mathcal{P}(B_2) + \pi \int B_2 \cup (B_2' + w_1^2) + \pi \int w_1^2 \cup B_2$$
$$+ \pi \int w_1 \cup \left[ (w_1 \cup (B_2' + w_1^2)) \cup_2 (B_2' + w_1^2) + w_1 \cup (B_2' + w_1^2) \right]$$
$$+ \frac{\pi}{2} \int w_1^2 \cup (B_2' + w_1^2), \tag{E.10}$$

For $B_2' = w_1^2$, where there is no 2-group symmetry, the anomaly is

$$\pi \int \mathcal{P}(B_2) + \pi \int w_1^2 \cup B_2. \tag{E.11}$$

For $B_2' = 0$, where there is 2-group symmetry, the anomaly is

$$\pi \int \mathcal{P}(B_2) + \pi \int w_1^2 \cup_1 w_1^3 + \pi \int w_1 \cup \left( w_1^3 \cup_2 w_1^2 \right) + \frac{3\pi}{2} \int w_1^4. \tag{E.12}$$

## E.4  $\mathbb{Z}_2^{\mathsf{T}}$ symmetry that does not permute anyons

We remark that we can also consider the case when the time-reversal symmetry does not act as charge conjugation. In such a case, we substitute $B_1 = 0$ instead of $B_1 = w_1$. Then $dB_2 = 0$ for all symmetry fractionalization classes, and the zero-form symmetry does not participate in a non-trivial two-group with the one-form symmetry. There are four fractional classes, labelled by $(n, m) \in \mathbb{Z}_2 \times \mathbb{Z}_2$, corresponding to $B_2 = n w_1^2$, $B_2' = m w_1^2$, with anomaly

$$n(m + 1)\pi \int w_1^4. \tag{E.13}$$

The class $n = 1, m = 0$ is the eTmT phase [119, 120], where electric and magnetic particles are both Kramers doublet.

---

[41]We note that in the case of two-group, one can still discuss different fractionalization classes by shifting the background gauge field, where background $B_2$ for the one-form symmetry and the background $B_1$ for the zero-form symmetry satisfies

$$d_\rho B_2 = B_1^* \Theta, \tag{E.8}$$

for Postnikov class $\Theta$. When $\Theta$ is non-trivial, the symmetry mixes to form a two-group, this implies that under a zero-form symmetry transformation with parameter $\lambda$, [21]

$$B_1 \to B_1^\lambda, \quad B_2 \to B_2 + \zeta(\lambda, B_1), \tag{E.9}$$

and this gives additional equivalence relation on the shift $B_2 \to B_2 + B_1^* \eta$ that implements different fractionalization classes [21].

# F   Two-groups for fractionalization classes of charge conjugation

In this appendix we discuss fractionalization classes for $\mathbb{Z}_2$ zero-form symmetry that acts as charge conjugation on $\mathbb{Z}_N$ or $\mathbb{Z}_2 \times \mathbb{Z}_2$ one-form symmetry. We will show that the zero-form charge conjugation symmetry with different fractionalization classes participates in different two-groups, for $\mathbb{Z}_N$ one-form symmetry with $N = 0 \bmod 4$ and $\mathbb{Z}_2 \times \mathbb{Z}_2$ one-form symmetry. The discussion applies to all spacetime dimensions.

## F.1   $\mathbb{Z}_N$ one-form symmetry

For odd $N$, there is only one fractionalization class. Therefore let us focus on even $N$, where there are two fractionalization classes for the charge conjugation zero-form symmetry. We will investigate whether some fractionalization class leads to zero-form symmetry that participates in a two-group.

When $N = 2 \bmod 4$, $N/2$ is odd, $\mathbb{Z}_N \cong \mathbb{Z}_{N/2} \times \mathbb{Z}_2$, and the charge conjugation symmetry acts non-trivially only on $\mathbb{Z}_{N/2}$, while the non-trivial fractionalization class lives in the other $\mathbb{Z}_2$. Thus changing the fractionalization classes of the zero-form symmetry does not make the symmetry participates in a two-group.

Let us turn to $N = 0 \bmod 4$, where $\mathbb{Z}_N$ contains $\mathbb{Z}_4$ subgroup. The charge conjugation symmetry sends $Q \in \mathbb{Z}_N$ to $-Q$. In particular, for $Q = N/4$ the difference is the non-trivial element in the $\mathbb{Z}_2$ subgroup. We can express the $\mathbb{Z}_N$ two-form gauge field for the one-form symmetry as $\mathbb{Z}_2$ gauge field $B$ and $\mathbb{Z}_{N/2} = \mathbb{Z}_N/\mathbb{Z}_2$ two-form gauge field $B'$. In the presence of the background $A$ for the charge conjugation symmetry, they obey

$$dB = \frac{dB'}{(N/2)} + B' \cup A \quad \bmod 2 . \tag{F.1}$$

This implies that the fractionalization class $B' = \frac{dA}{2} \bmod N/2$ gives zero-form symmetry that participates in a two-group with the $\mathbb{Z}_2$ subgroup one-form symmetry invariant under the charge conjugation symmetry,

$$dB = A \cup \frac{dA}{2} \quad \bmod 2 , \tag{F.2}$$

where by suitable redefinition of $B$ this is the same as $dB = A^3 \bmod 2$.

We remark that another fractionalization class that corresponds to shifting $B \to B + dA/2$ does not give rise to a two-group, since $dB$ is not affected by the shift.

## F.2   $\mathbb{Z}_2 \times \mathbb{Z}_2$ one-form symmetry

Consider $\mathbb{Z}_2 \times \mathbb{Z}_2$ one-form symmetry with the charge conjugation exchanging the two $\mathbb{Z}_2$'s. Denote the background two-form gauge field for the diagonal $\mathbb{Z}_2$ generator by $B$, and the background for one of the $\mathbb{Z}_2$ generator by $B'$, and the background for the charge conjugation symmetry by $A$, then they satisfy

$$dB = B' \cup A \bmod 2 , \quad dB' = 0 \bmod 2 . \tag{F.3}$$

This implies that the fractionalization class $B' = A \cup A$ gives zero-form symmetry that participates in a two-group with the $\mathbb{Z}_2$ subgroup one-form symmetry invariant under charge conjugation symmetry,

$$dB = A^3 \bmod 2 . \tag{F.4}$$

# G  Twisted gauge bundle and two-group symmetry

Let us consider a gauge theory with simply-connected gauge group $G_{\text{gauge}}$ and matter fields that transform under global symmetry $\tilde{G}_{\text{global}}$ which commutes with the gauge group (and in particular, does not act on the center one-form symmetry). Denote by $\Gamma$ the center subgroup of the gauge group that acts trivially on the matter fields, which gives the one-form symmetry in the gauge theory. The action of a center subgroup $\mathcal{Z} \subset \tilde{G}_{\text{global}}$ is identified with a transformation in the center of the gauge group. The symmetry of the classical action is

$$\frac{(G_{\text{gauge}}/\Gamma) \times \tilde{G}_{\text{global}}}{\mathcal{Z}}\,. \tag{G.1}$$

We note that $\mathcal{Z}$ in the gauge group participates in a central extension

$$1 \to \Gamma \to \tilde{\mathcal{Z}} \to \mathcal{Z} \to 1\,, \tag{G.2}$$

where $\tilde{\mathcal{Z}}/\Gamma = \mathcal{Z}$, and $\tilde{\mathcal{Z}}$ is a subgroup in the center of $G_{\text{gauge}}$ that is larger than $\Gamma$. When we turn on a background two-form gauge field $B_2$ for the $\Gamma$ one-form symmetry and the background for the $G_{\text{global}} = \tilde{G}_{\text{global}}/\mathcal{Z}$ zero-form symmetry, the gauge bundle is twisted to be a $G_{\text{gauge}}/\tilde{\mathcal{Z}}$ bundle. The background for $G_{\text{global}}$ symmetry can be described by an obstruction class $w_2^G$ to lifting the bundle to a $\tilde{G}_{\text{global}}$, and it is a two-form that takes value in $\mathcal{Z}$. Concretely, on triple overlaps of coordinate patches, the transition functions that take value in $\tilde{G}_{\text{global}}$ only multiply to an element in $\mathcal{Z} \subset \tilde{G}_{\text{global}}$ for $G_{\text{global}} = \tilde{G}_{\text{global}}/\mathcal{Z}$ bundles, and the equivalence class of the element under redefinition of each transition function by elements in $\mathcal{Z}$ gives $w_2^G$. The two-forms $w_2^G$ and $B_2$ combine into a two-form gauge field for $\tilde{\mathcal{Z}}$, and this implies that they obey the constraint

$$dB = \text{Bock}\left(w_2^G\right)\,, \tag{G.3}$$

where Bock is the Bockstein homomorphism for the short exact sequence (G.2). This implies that if $w_2^G$, which is a $\mathcal{Z}$-valued two-cocycle, cannot be lifted into a $\tilde{\mathcal{Z}}$-valued two-cocycle, then the zero-form symmetry combines with the one-form symmetry into a two-group symmetry, where the Postnikov class that characterized the two-group is [21, 121]

$$\Theta_3 = \text{Bock}(w_2^G) \in H^3\left(BG_{\text{global}}, \Gamma\right)\,. \tag{G.4}$$

Examples of such theories with two-group symmetry are discussed in [21, 121].

In particular, consider the case $G_{\text{global}} = \mathbb{Z}_n$, $\Gamma = \mathbb{Z}_N$ and $\tilde{G}_{\text{global}}$ is a central extension of $G_{\text{global}}$ by $\mathcal{Z} = \mathbb{Z}_m$, then

$$w_2^G = r\frac{dA}{n} \bmod m\,, \tag{G.5}$$

where $r$ is an integer that specifies the extension $\tilde{G}_{\text{global}}$, $A$ is the $G_{\text{global}}$ background gauge field. Then

$$\text{Bock}(w_2^G) = r'\frac{dw_2^G}{m} \bmod N = 0\,, \tag{G.6}$$

where $r'$ is an integer that specifies the extension $\tilde{\mathcal{Z}}$ of $\mathcal{Z}$ by $\Gamma$. Thus such zero-form symmetry does not participate in a two-group.

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
