# Peer review of "Anomalies and Symmetry Fractionalization"

_SciPost Physics, doi:SciPost Phys. 15, 079 (2023)_

## Round 2 · Referee Report · Anonymous (Referee 1) · 2023-5-31

Report

This article examined theories in which both a zero-form and a one-form global symmetries coexist. The main message is as follows. The fact that charged line operators may transform under a projective representation of the zero-form global symmetry leads to an ambiguity of its action on the former, and additional discrete data must be supplemented in order to unambiguously determine such an action on the line defects. This phenomenon is referred to as the symmetry fractionalization, and the discrete information is known as a fractionalization class. A far-reaching consequence is that the ’t Hooft anomaly for the zero-form symmetry may be affected by this phenomenon. Moreover, if one twists the action of the zero-form global symmetry by a gauge transformation, this can lead to a change of fractionalization class and may result in the jump in the 't Hooft anomaly of the zero-form symmetry.

Although this phenomenon had been known in the condensed matter literature, this paper formulated the problem in the contemporary high-energy physics language. One of the strong points of this article is that several instructive physical examples in various spacetime dimensions were provided, and the results were explained from many points of view. Note that there have already been several improvements in the presentation as well as explanations in Version 2 of this article on the arXiv. Given that this work contains several important results and will serve as an important reference on the subject, the referee recommends it for publication without any hesitation.

---

## Round 2 · Referee Report · Anonymous (Referee 2) · 2023-6-2

Strengths

The paper provides a clear introduction to the concept of symmetry fractionalization and its effect on anomaly calculations. The examples are also well-chosen and pleasantly increase in complexity step by step.

Report

This is an excellent paper, simultaneously novel, thorough, and pedagogical. It is clear the authors put a lot of thought and effort into the problem and its presentation. I definitely recommend the paper for publication, and I will also be recommending it for others to read to learn about symmetry fractionalization and anomaly calculations in gauge theories and other theories with 1-form symmetries.

There is just one technical comment I would like to make, which the authors certainly already understand, concerning a slightly different way of looking at the ambiguity in the 0-form anomaly the authors present. That is, we can consider a theory with a 1-form symmetry \Gamma as having an action by the 2-group B\Gamma (B here just indicates that we put the abelian group \Gamma in the 1-form degree. It is meant to evoke the notation of the classifying space, considered as an H-space.). Suppose there is also a 0-form symmetry G, and no Postnikov class or twisting action, so that the total symmetry is a product 2-group G \times B\Gamma.

The ambiguity discussed in the paper can be understood in terms of 2-group automorphisms of G \times B\Gamma. In particular, we have the "T type" (outer) automorphisms which act as the identity on B\Gamma, and project to the identity over G, determined by a map G \to B\Gamma. Such maps are classified by \eta \in H^2(G,\Gamma) and these automorphisms can be thought of as twists of the symmetry data. The action of this automorphism on the G and \Gamma gauge fields A and B (resp.) can be written

A \mapsto A
B \mapsto B + \eta(A)

This is just like the background field-dependent gauge transformation considered by the authors, e.g. near equation 2.37. We see the ambiguity in the 't Hooft anomaly is similar to that when we have outer automorphisms of G.

---

## Round 3 · Author Response

We thank the referees for the comments. We added footnote 23 as suggested by the second referee in the report 2.

---

## Round 3 · List of Changes

Added footnote 23 on p20 as suggested in the second referee report.

---

## Editorial Decision

published